# Age-related changes in polycomb gene regulation disrupt lineage fidelity in intestinal stem cells

Helen M Tauc[1], Imilce A Rodriguez-Fernandez[1], Jason A Hackney[2], Michal Pawlak[3], Tal Ronnen Oron[4], Jerome Korzelius[5], Hagar F Moussa[6], Subhra Chaudhuri[7], Zora Modrusan[1,7], Bruce A Edgar[8], Heinrich Jasper[1]*

[1]Immunology Discovery, Genentech, South San Francisco, United States; [2]OMNI Bioinformatics, Genentech, South San Francisco, United States; [3]Institute of Hematology and Blood Transfusion, Warsaw, Poland; [4]Buck Institute for Research on Aging, Novato, United States; [5]School of Biosciences, University of Kent, Canterbury, United Kingdom; [6]Department of Biomedical Engineering and Biological Design Center,Boston University, Boston, United States; [7]Department of Microchemistry, Proteomics, Lipidomics and Next Generation Sequencing, Genentech, South San Francisco, United States; [8]Huntsman Cancer Institute, University of Utah, Salt Lake City, United States

**Abstract** Tissue homeostasis requires long-term lineage fidelity of somatic stem cells. Whether and how age-related changes in somatic stem cells impact the faithful execution of lineage decisions remains largely unknown. Here, we address this question using genome-wide chromatin accessibility and transcriptome analysis as well as single-cell RNA-seq to explore stem-cell-intrinsic changes in the aging *Drosophila* intestine. These studies indicate that in stem cells of old flies, promoters of Polycomb (Pc) target genes become differentially accessible, resulting in the increased expression of enteroendocrine (EE) cell specification genes. Consistently, we find age-related changes in the composition of the EE progenitor cell population in aging intestines, as well as a significant increase in the proportion of EE-specified intestinal stem cells (ISCs) and progenitors in aging flies. We further confirm that Pc-mediated chromatin regulation is a critical determinant of EE cell specification in the *Drosophila* intestine. Pc is required to maintain expression of stem cell genes while ensuring repression of differentiation and specification genes. Our results identify Pc group proteins as central regulators of lineage identity in the intestinal epithelium and highlight the impact of age-related decline in chromatin regulation on tissue homeostasis.

*For correspondence:
jasper.heinrich@gene.com

## Introduction

Somatic stem cells (SCs) are critical for tissue regeneration and maintenance as they have the unique ability to self-renew as well as to generate all cell types of the tissue in which they reside. Maintaining homeostasis requires SCs to respond appropriately to the needs of the tissue and generate pertinent cell types when necessary. SCs must therefore uphold a state of plasticity that allows for dynamic and appropriate lineage commitment. Transcriptional and chromatin landscaping studies have suggested that an interplay of genetic and epigenetic mechanisms defines and ensures SC lineage fidelity in various SC systems (*Adam and Fuchs, 2016*). In hair follicle (HF) and hematopoietic SCs (HSCs), master transcriptional regulators dictate lineage choice by controlling chromatin state and transcriptional networks driving differentiation and lineage restriction (*Adam et al., 2018*; *Lara-Astiaso et al., 2014*), while in the mouse intestinal epithelium, lineage-specific transcription factors

define early cell fate decisions in a relatively permissive epigenetic landscape (*Jadhav et al., 2017*; *Kim et al., 2014*). While precise mechanisms of lineage specification may differ, a common problem across aging tissues is that, in addition to stem cell regenerative capabilities, lineage fidelity declines. HSCs, for example, not only lose their ability for long-term reconstitution in transplantation assays (*Kamminga et al., 2005*; *Rossi et al., 2005*), but also become skewed toward the myeloid lineage in aging mice (*Elias et al., 2017*).

The functional decline in SCs over time is correlated with, and may be caused by, aberrant epigenetic regulation (*Beerman and Rossi, 2015*; *Ermolaeva et al., 2018*; *Krauss and de Haan, 2016*). The universality and mechanism of these age-related epigenetic changes in SCs, and how critical they are for age-related regenerative decline, remain largely unclear. Chromatin regulation at the nucleosome level includes post-translational modifications (PTMs) of histone tails that coordinate recruitment of chromatin modifiers, affecting nucleosome spacing and DNA accessibility (*Bannister and Kouzarides, 2011*). A major class of chromatin regulators are the Polycomb Group (PcG) and Trithorax Group (TrxG) proteins, classically characterized as mediators of silent or active chromatin, respectively (*Geisler and Paro, 2015*). Polycomb Repressor Complex 2 (PRC2) is responsible for depositing methyl moieties onto H3K27, rendering it a repressive mark, and Polycomb Repressor Complex 1 (PRC1) recognizes and binds H3K27me3 and monoubiquitinates histone H2A on lysine K119 (*Schuettengruber et al., 2017*). TrxG complexes maintain a transcriptionally active state by promoting H3K4 methylation and H3K27 acetylation, but their exact composition is context dependent (*Geisler and Paro, 2015*).

PRC1 and PRC2 are well studied in the context of development and play important roles in lineage determination and restriction (*Brand et al., 2019*). Yet it is becoming clear that PcG proteins also have post-developmental cell-type-specific roles in various cells and tissues, including in adult SC regulation. PcG proteins regulate SC identity and function in the hematopoietic system as well as in the muscle, intestine, skin, and brain (*Avgustinova and Benitah, 2016*; *Beerman and Rossi, 2015*; *Ezhkova et al., 2011*; *Hidalgo et al., 2012*; *Iwama et al., 2004*; *Jacobs et al., 1999*; *Juan et al., 2011*; *Kamminga et al., 2006*; *Kinkel et al., 2015*; *Oguro et al., 2010*; *Pivetti et al., 2019*; *Robson et al., 2011*; *Shen et al., 2008*; *Sousa-Victor et al., 2014*; *Woodhouse et al., 2013*; *Xie et al., 2014*; *Zhang et al., 2014*). In the mouse intestine, PRC1 ensures intestinal SC (ISC) identity by repressing non-intestinal lineage transcription factor genes, which, if activated, interfere with the transcriptional response downstream of Wnt signaling, resulting in loss of ISC self-renewal (*Chiacchiera et al., 2016a*). PRC2, in turn, was shown to be dispensable for ISC maintenance, but required for progenitor cell proliferation and differentiation (*Chiacchiera et al., 2016b*). In this lineage, the PRC1 component Bmi1 further marks a reserve stem cell population (*Yan et al., 2012*) that is committed to the enteroendocrine (EE) cell fate (*Yan et al., 2017*). These cells can de-differentiate and repopulate the basal crypt after depletion of Lgr5+ intestinal stem cells (*Jadhav et al., 2017*). How aging impacts the function of PcG proteins in SC regulation remains unknown, but similar derepression of p16INK4a has been implicated in the decline of regenerative capacity of geriatric as well as Bmi1-deficient muscle SCs (MuSCs), suggesting that age-related changes in PcG protein function play critical roles in SC aging (*Sousa-Victor et al., 2014*).

A few studies have examined global changes in histone modifications specifically in SCs during aging (*Krauss and de Haan, 2016*). Broadening of H3K4me3 marks at identity and self-renewal genes and increased DNA methylation on differentiation genes was observed in old HSCs (*Sun et al., 2014*), and increased H3K27me3 in old MuSCs was postulated to be linked to their dysfunction (*Liu et al., 2013*). These studies support the idea that during aging, a redistribution of epigenetic marks causes misregulation of transcriptional programs critical for SC maintenance and lineage potency. However, the extent of these changes in SCs, as well as the mechanism(s) by which these changes impact SC function remain to be fully elucidated.

The *Drosophila* intestine serves as a powerful model in which to investigate regulatory principles for stem cell function and the age-related decline of such mechanisms. The midgut epithelium is maintained and regenerated by resident ISCs, which give rise to both the enterocyte (EC) and the EE cell lineages (*Micchelli and Perrimon, 2006*; *Ohlstein and Spradling, 2006*). Lineage commitment into either the EC or EE lineage is largely controlled by Notch (N) signaling, where high N promotes the EC lineage through a transient post-mitotic progenitor called the enteroblast (EB), and low N activity is associated with EE differentiation (*Micchelli and Perrimon, 2006*; *Ohlstein and Spradling, 2006*).

Intestinal turnover in young flies is relatively slow, and ISCs thus reside largely in a quiescent state. In aging flies, however, ISCs become hyper-proliferative, due to increased stress signaling linked to commensal dysbiosis and the epithelial inflammatory response. EC differentiation further becomes misregulated, in part due to aberrant N signaling. Together, these changes result in epithelial dysplasia and barrier dysfunction (*Biteau et al., 2008*; *Biteau et al., 2010*; *Rera et al., 2012*). Age-related changes in ISC activity driven by both intrinsic and environmental influences have been studied extensively (*Jasper, 2020*). It has not been examined, however, whether and how changes in ISC lineage fidelity contribute to age-related pathologies in this model.

Here, we used the *Drosophila* intestine to investigate age-related changes in ISC lineage fidelity. We find that in aging flies, ISCs exhibit transcriptomic and chromatin accessibility changes that prime them toward the EE lineage. Accordingly, single-cell RNA-seq (scRNA-seq) and histological analysis uncover an age-associated increase in the proportion of EE cells as well as EE-specified ISCs. We find that these changes are mediated by deregulation of Polycomb (Pc) target genes, and that aging ISCs acquire H3K27me2 marks that are reminiscent of young EEs. We further show that Pc and trithorax (trx) are central regulators of ISC identity and EE lineage commitment, maintaining a balanced antagonism critical for appropriate expression of lineage-specific and stem cell genes. Finally, we find that the Pc-mediated increase in EE cell numbers is a consequence of commensal dysbiosis-mediated JNK activation in ISCs, and that it contributes to the age-related dysplasia characteristic of the old *Drosophila* intestine. Taken together, our results uncover PcG proteins as key regulators of lineage identity and fidelity in the ISC lineage, and identify the perturbation of PcG gene regulation as a major driver of the age-related loss in tissue homeostasis.

## Results

### Chromatin accessibility and transcriptome analysis on aging ISCs

While a large number of studies have explored systemic changes and changes in the local microenvironment that occur with age and impact the function of ISCs in *Drosophila*, comparably little is known about the intrinsic changes to ISC identity, function and regulation that may influence ISCs function in the aging animal (*Jasper, 2020*). To start exploring such changes, we performed ATAC-seq and RNA-seq on fluorescent-activated cell sorting (FACS)-purified ISCs isolated from flies across their lifespan, covering early adulthood (7 days of age), middle age (30 days of age), old age (60 days), and geriatric age (>70 days). To minimize genetic bias due to inbreeding, ISCs were isolated from an outcross of esg-Gal4, UAS-2xYFP; Su(H)-Gal80 to $W^{Dah}$ wild-type animals. In these animals, ISCs were specifically labeled by YFP, as esg-Gal4 drives expression in ISCs and EBs, yet Su(H)-Gal80 represses expression in EBs (*Wang et al., 2014*). For each biological replicate, a cohort of ~120 flies were dissected, the intestinal tissue was dissociated into single cells, and YFP+ ISCs were isolated by FACS using previously described methods (*Tauc et al., 2014*; *Figure 1A and B*).

Chromatin accessibility, as assessed by ATAC-seq, revealed an overall stable global profile of accessible sites across all ages (*Figure 1C and D*, *Figure 1—figure supplement 1C and E*). We noted a very mild, but reproducible, relaxation of chromatin, apparent as an increase in accessibility, in mid-aged ISCs, followed by a decrease in accessibility at 60 days (*Figure 1C*, *Figure 1—figure supplement 1C*). Pearson correlation analysis confirmed that all samples were highly similar to each other and that accessibility differences between young and old animals were mild (*Figure 1D*). This suggests that ISCs maintain a stable and robust chromatin accessibility profile along the whole lifespan of the animal. Accordingly, chromatin accessibility around selected ISC marker genes like *esg* remained unchanged at all time-points (*Figure 1—figure supplement 1F*). Principal component analysis (PCA) of peak accessibility, however, revealed a separation between young samples and mid-age to geriatric samples (*Figure 1—figure supplement 1A and D*), and we were able to identify a set of differentially accessible peaks between old and young samples (*Figure 1E*). These were primarily located in promoters, introns and intergenic regions (*Figure 1—figure supplement 1B*). Focusing on the differentially accessible promoter regions (*Figure 1E* and *Supplementary file 1*), we performed a motif analysis on each age comparison and noted a consistent enrichment for Trithorax-like (Trl) and Zeste (z) motifs across all time-point comparisons (*Figure 1F*). Trl and z have been reported to bind to Polycomb Repressor Elements (PREs) and recruit PcG proteins to PREs

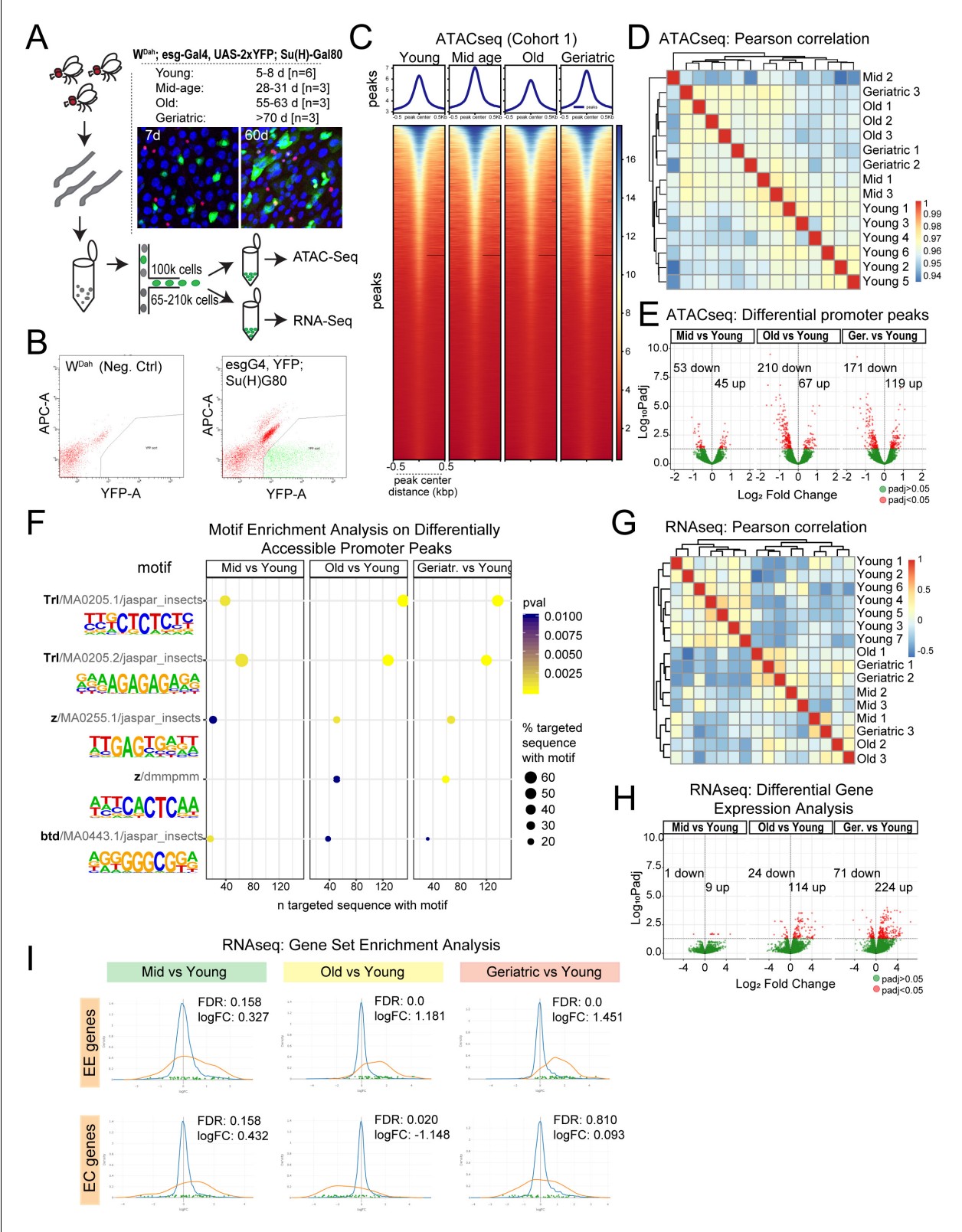

**Figure 1.** Chromatin accessibility and transcriptional changes in aging ISCs suggest a perturbation in lineage fidelity. (**A**) Experimental design of midgut dissection and ISC isolation. Inset shows an example of a young and aged midgut of the fly line used with ISCs marked by YFP (green), EEs marked by Prospero (red), and DAPI in blue. n represents the number of samples containing ISCs from ~120 guts of the indicated age. (**B**) Example of the gating strategy used in Fluorescence Activated Cell Sorting (FACS) for isolation of YFP+ ISCs. (**C**) Heatmap of average chromatin accessibility in

*Figure 1 continued on next page*

*Figure 1 continued*

ISCs at young, mid-age, old, and geriatric time-points. (D) Pearson correlation heatmap showing ATAC-seq sample similarity. (E) Volcano plots of significantly differentially accessible peaks at gene promoter regions at aged time-points compared to young. See also *Supplementary file 1*. (F) Motif analysis of differentially accessible promoter regions. (G) Pearson correlation heatmap showing RNA-seq sample similarity. (H) Volcano plots of significantly differentially regulated genes. See also *Supplementary file 2*. (I) Gene Set Enrichment Analysis of an EE and EC geneset at aging time-point comparisons. Figure supplement 1 and 2. contain additional comparisons of sample similarity of aging ISCs from ATAC-seq and RNA-seq experiments.

The online version of this article includes the following figure supplement(s) for figure 1:

**Figure supplement 1.** Additional comparisons of sample similarity of aging ISCs from ATAC-seq and RNA-seq experiments.
**Figure supplement 2.** Correlation between gene expression and promoter accessibility.

(*Müller and Kassis, 2006*). The differential chromatin accessibility at Trl and z sites thus suggests a potential mis-regulation of PcG proteins in old ISCs.

To gain insight into whether the differential chromatin accessibility observed translates into differential gene expression in aging ISCs, we performed RNA-seq on ISCs that were isolated at the same time-points as the samples for ATAC-seq. When all differentially expressed genes were compared with all differentially open chromatin peaks, the correlation was low (*Figure 1—figure supplement 2A*). However, a strong positive correlation was observed for genes that were significantly (p<0.05) changed in both transcription (*Supplementary file 2*) and promoter accessibility (*Figure 1—figure supplement 2B*). Of the genes that exhibited changes at both levels, 58% (18/31 genes) contained a Trl and z motif (*Supplementary file 3*).

Pearson correlation of RNA-seq samples showed that young samples clustered separately from mid-aged to geriatric samples, suggesting that ISCs begin to exhibit distinct transcriptional differences at mid-age (*Figure 1G*). Accordingly, young samples separated from mid- to geriatric samples in PCA analysis (*Figure 1—figure supplement 1G*). Compared to young time-points, mid-age time-points exhibited significant differential expression (adjusted p-value<0.05) for only a few genes (*Figure 1H*), but there was a significant increase in differentially regulated genes at old and geriatric time-points (*Figure 1H*), suggesting a progressive change in gene expression of ISCs as the animal ages. Focusing on genes with a $\log_2$ fold change (logFC) of +/- 1, we found no gene ontology (GO) or pathway enrichment for genes differentially regulated at mid and old time-points. Deregulated genes in geriatric samples were enriched for glutathione metabolic process [GO:0006749] as well as genes in the drug metabolism -cytochrome P450 pathway. We also noticed an upregulation of neuronal genes that are also markers of EE cells such as *Sytalpha*, *nrv3* and *dysc* in both old and geriatric samples (*Supplementary file 2*). An upregulation of neuronal genes in old ISCs is in accordance with results from a previous study (*Tauc et al., 2017*). Accordingly, Gene Set Enrichment Analysis (GSEA) with gene sets of 58 or 51 previously described EE or EC-specific genes, respectively (*Dutta et al., 2015*), revealed a consistent and significant enrichment of EE genes in the genes upregulated in aging ISCs (*Figure 1I*). EC genes were downregulated at the old time-point, and no significant trends were observed for these genes in other comparisons (*Figure 1I*). These results suggested an age-related increase in the acquisition of EE fates at the expense of EC fates. To test this hypothesis, we performed scRNA-seq and immunohistochemistry to explore lineage dynamics in the aging intestinal epithelium.

## Single-cell RNA-seq on whole aging midguts

We performed scRNA-seq on whole midguts at young, mid-age, and old time-points and a total of 4880 young, 1440 mid-aged, and 5829 old cells were used in our final analyses. After batch correction, clustering analysis revealed nine clusters that could be classified into the major cell types and sub-cell types of the midgut that have been recently described (*Hung et al., 2020*; *Figure 2A*, *Figure 2—figure supplement 1*). Focusing on age-related changes in the ISC/EB populations, we confirmed the well-characterized (*Jasper, 2020*) age-associated increase in mitotically active ISCs, which separated into a distinct cluster (#3) that consisted of cells with elevated expression of cell cycle and mitotic genes (*Figure 2A and B*, *Figure 2—figure supplement 1*). Two cell clusters (#4 and 8) were almost exclusively present in samples from old flies (*Figure 2A and B*). The transcriptomes of cells from these clusters were enriched for transcripts encoding genes involved in DNA repair, TNFalpha/

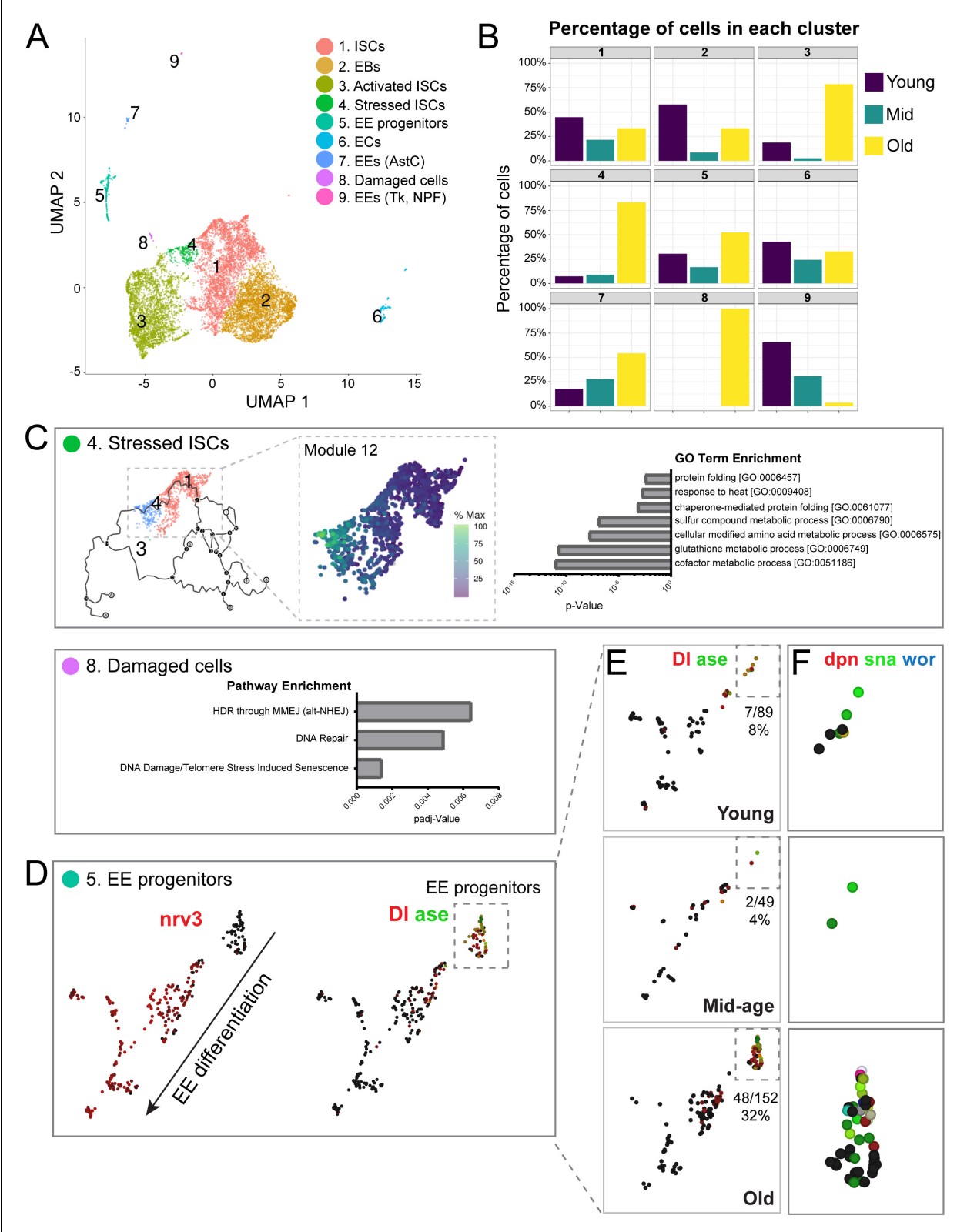

**Figure 2.** Single-cell RNA-seq analysis reveals old 'stressed' ISCs and an increase in EE progenitors during aging. (**A**) Uniform Manifold Approximation and Projection (UMAP) plot of combined scRNA-seq data including cells of young, mid-age, and old midguts showing cluster annotation. Clusters were annotated based on gene set enrichment analyses (*Figure 2—figure supplement 1*). Genetic changes observed in our scRNA-seq were also compared to our bulk RNA-seq data to ensure robustness of results (*Figure 2—figure supplements 2* and *3*). (**B**) Age composition of each cluster showing the
*Figure 2 continued on next page*

*Figure 2 continued*

proportion of cells from young, mid-age and old samples. (**C**) Top panel: (left) pseudotime of cells in the UMAP (above in A) focusing on the trajectory of young cells (Cluster 1) to old cells (Cluster 4). (middle) Differentially expressed genes along the trajectory (*Figure 2—figure supplement 4*), focusing on genes that become upregulated in Cluster 4 (Module 12, *Figure 2—figure supplement 5*). Cells are colored by relative scaling of gene expression to the maximum expression of a given gene. (right) GO term enrichment analysis (Module 12 genes) is shown in the bar graph. Bottom panel: Bar graph showing pathway enrichment for genes expressed in Cluster 8, Module 18 (*Figure 2—figure supplement 4*). (**D**) Detailed analysis of Cluster five showing gene expression of mature EE marker *nrv3* (left) and EE progenitor markers *Dl* and *ase* (right; high expression in red or green, lack of expression in black). Note the direction of differentiation and location of EE progenitors (boxed with dotted line). *Figure 2—figure supplement 6* shows additional expression patterns for EE and EE progenitor genes. (**E**) Split view of Cluster five in D (image on the right) based on age. EE progenitor cells (expressing *Dl* and *ase*) are in the outlined box and the proportion of those cells within Cluster five at each age is indicated below each box. (**F**) Close-up of EE progenitors showing upregulation of pro-neural genes *dpn*, *sna*, and *wor* in red, green, and blue, respectively (see also *Figure 2—figure supplement 7*). Black indicates no expression.

The online version of this article includes the following figure supplement(s) for figure 2:

**Figure supplement 1.** Gene set enrichment analysis per single-cell cluster.
**Figure supplement 2.** Comparison of genetic changes during aging in scRNA-seq ISCs with bulk RNA-seq.
**Figure supplement 3.** Differentially regulated genesets from bulk RNAseq are enriched in scRNA-seq analysis.
**Figure supplement 4.** Pseudotime and trajectory analysis of single-cell RNA-seq data from aging ISCs.
**Figure supplement 5.** Gene trajectory analysis of Cluster 4.
**Figure supplement 6.** Gene trajectory analysis of Cluster 5.
**Figure supplement 7.** Neural stem cell gene signature is upregulated in old ISCs.

NFkB signaling and protein secretion, while transcripts involved in the cell cycle and mitosis were depleted or downregulated compared to other clusters (*Figure 2—figure supplement 1*).

To explore which clusters contribute most to the genetic changes identified in our bulk RNA-seq, we performed a 'pseudo bulk' analysis on individual clusters (comparing mid-age to young and old to young cells) and performed GSEA for genes differentially regulated in our bulk RNA-seq during aging. We observed a significant correlation of differentially expressed genes when comparing young with mid-aged and young with old samples (correlation is weaker in the second comparison, potentially due to an age-related increase of transcriptional noise, *Figure 2—figure supplement 2*). We also found that age-associated genetic changes in Clusters 1–3 showed the most significant enrichment and gene expression overlap with our bulk RNA-seq gene sets (*Figure 2—figure supplement 3*). These results indicate that transcriptional changes in our bulk RNA-seq are not unique to a particular cluster, but rather are upregulated in all ISC and progenitor clusters (Clusters #1–3). We noted that in old samples, a substantial number of genes were also upregulated in Cluster #4 (consisting of mainly old cells) and #5 (EE progenitor cells). Overall, these results show that the bulk and single cell RNA-seq analysis reveal consistent changes in the aging ISCs populations in a complementary manner.

We next sought to explore differences between and within ISC clusters more specifically. Clusters #4 and #8, which are made up almost exclusively of old cells, exhibited similar GO term enrichment (*Figure 2—figure supplement 1*). To understand the biology contributing to the separation of these two clusters better, we used Monocle3 (*Cao et al., 2019*; *Qiu et al., 2017*; *Trapnell et al., 2014*) to order cells in pseudotime and performed trajectory analyses to identify gene modules driving cluster identities (*Figure 2—figure supplement 4*). Cluster 4 trajectory analysis identified gene modules whose expression best represent the progressive distancing of Cluster 1 cells from Cluster 4 cells (i.e. from young to old ISCs) (*Figure 2C*, *Figure 2—figure supplements 4* and *5*). GO analysis on gene module 12 (*Figure 2—figure supplement 5*) revealed enrichment of stress-associated terms such as 'glutathione metabolic process', 'chaperone-mediated protein folding' and 'response to heat' (*Figure 2C*), suggesting that Cluster #4 consists of stressed cells. Of note, we found that many significantly upregulated genes from the bulk RNA-seq were also upregulated in Cluster #4 (*Figure 2—figure supplement 5*, *Supplementary file 4*), suggesting that the most pronounced expression changes observed in bulk ISCs reflect the acquisition of the 'aged/stressed ISC' state.

In contrast, Cluster #8 is disconnected from the trajectory root of young ISCs, and we used gene sets significantly enriched in the module analysis to characterize these cells (*Figure 2—figure supplement 4B*). Module 18 genes are upregulated in Cluster #8 (*Figure 2—figure supplement 4B*) and show enrichment for DNA repair, DNA damage and telomere stress induced senescence

pathways (*Figure 2C*), indicating damaged cells. The appearance of these 'stressed' or damaged cell states in old intestines is consistent with reported elevated stress signaling as well as DNA damage in the aged epithelium (*Buchon et al., 2009*; *Guo et al., 2014*; *Hochmuth et al., 2011*; *Park et al., 2012*; *Siudeja et al., 2015*). During aging, loss of activity of the antioxidant response regulator, CncC/Nrf2, leads to an accumulation of reactive oxygen species (ROS) in old ISCs (*Hochmuth et al., 2011*). High levels of ROS have been linked to increased DNA damage in these cells, an observation that is supported by our results (*Park et al., 2012*).

While the emergence of Clusters #4 and #8 were thus consistent with previous work, we were intrigued by the unexpected age-related increase in EE signatures in ISCs observed in the bulk RNA-seq analysis. To explore possible changes in lineage specification further, we performed gene expression trajectory analysis for Cluster #5 as it consists of EE progenitor cells. This revealed the progression of EE differentiation from EE-specified (Pros+/esg+) ISC progenitors to mature EEs, and included mitotic EE progenitors, which most likely represent a mitotic phase during EE specification that has recently been described (*Chen et al., 2018*; *He et al., 2018*; *Figure 2—figure supplement 6*). Focusing on the EE progenitor sub-cluster (based on *Delta* (*Dl*) and *asense* (*ase*) expression and lack of *nervana 3* (*nrv3*), a mature EE marker), we observed a significant increase in the number of EE progenitor cells in old samples (*Figure 2E*). Strikingly, old EE progenitors had increased expression of *deadpan* (*dpn*) and other pro-neural genes (*Figure 2F*), the upregulation of which was recently described as a hallmark of neuroendocrine tumor stem cells in the midgut (*Li et al., 2020*). We also noted a general upregulation of these neural stem cell (NSC) markers in our bulk RNA-seq (*Figure 2—figure supplement 7*). The significance of this acquisition of an NSC- like phenotype in the aging intestine is intriguing and will require further exploration. Critically, our scRNA-seq data revealed that the upregulation of EE genes in aged ISCs observed by bulk RNA-seq was driven by an increase in the number of EE-specified ISCs. We decided to explore the origin of these cells and the impact of their accumulation on tissue homeostasis.

## Increase in EE cell lineage in old posterior midguts

We first confirmed the age-related skewing of the ISC lineage using immunohistochemistry to quantify the proportion of EEs in posterior midguts. We used either a genetic reporter, based on the EE marker gene Prospero (*Pros*$^{V1}$*-Gal4, UAS-GFP Grosjean et al., 2001*), or detected Pros+ cells directly using immunostaining in intestines of flies expressing YFP specifically in ISCs (*W*$^{Dah}$*;esg-Gal4, UAS-2xYFP; Su(H)-Gal80*). Both methods showed a significant and progressive increase in the proportion of EE cells as the animal ages (*Figure 3A,B*). We also observed an increase in esg and Pros double-positive cells in old midguts (*Figure 3B*), indicating an increase in EE progenitor cells. These analyses were performed on posterior midguts (R4 and R5 regions; *Buchon et al., 2013*).

## Old midguts, including ISCs, accumulate H3K27me2

The enrichment for Trl and z motifs in differentially open promoters in old ISCs based on our ATAC-seq analysis, as well as an upregulation of classic Pc targets (*Abd-B* and *Ubx*) in our bulk RNA-seq data (*Supplementary file 2*), strongly suggested an involvement of PcG proteins in the age-related loss of lineage fidelity. Both Trl and z are DNA-binding proteins that can bind PREs and have been implicated in cooperating with PcG complexes to regulate gene repression (*Erokhin et al., 2018*; *Mulholland et al., 2003*). Since PRC2 is responsible for depositing methyl groups onto H3K27 (*Laugesen et al., 2016*; *Margueron and Reinberg, 2011*), we asked whether aging affects H3K27 methylation in intestinal cells of the fly. To this end, Luminex xMAP technology (Active Motif) was used to quantify levels of H3K27me2 and H3K27me3 alongside a panel of other post-translational histone modifications in whole midgut tissue from young and old flies (*Figure 4—figure supplement 1*). This analysis revealed a 153% increase in H3K27me2 as well as a 65% increase in pan-acetylated H3 in old flies (*Figure 4—figure supplement 1*). Antibody staining for H3K27me2 confirmed that H3K27me2 levels change in older intestines, albeit in a cell-type-specific manner: ISC nuclei exhibited a lower intensity of H3K27me2 than EEs in young animals (*Figure 4A*), however, ISCs in old midguts exhibited a significant increase in H3K27me2 intensity, approaching EE levels (*Figure 4A*). This accumulation of H3K27me2 in aging ISCs thus further supported the notion that as ISCs age, they acquire an epigenetic state resembling EEs.

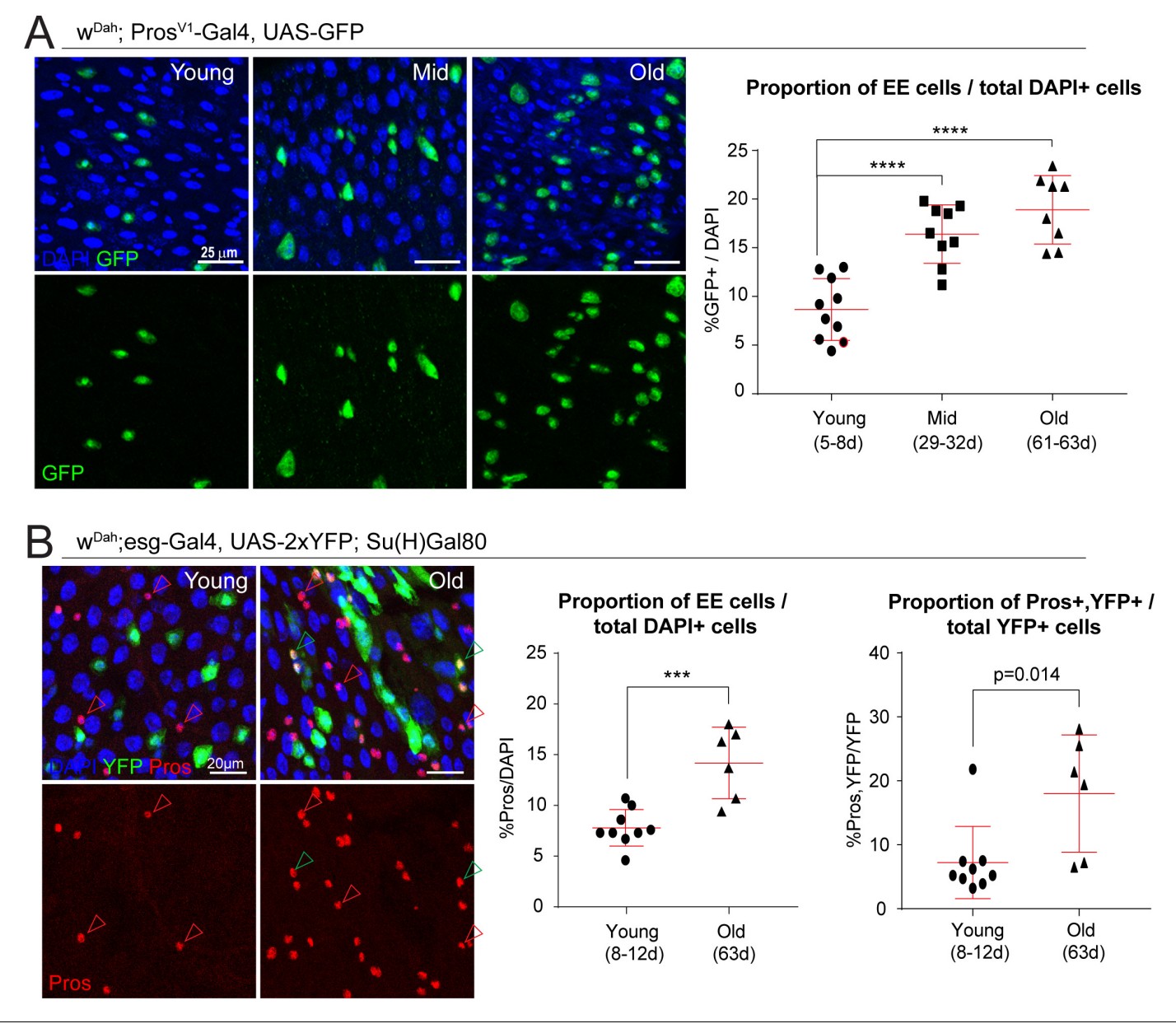

**Figure 3.** The proportion of EE cells and EE progenitors increases during aging. (**A**) Confocal images showing fluorescent immunohistochemistry on posterior midguts of young, mid-age, and old flies (ages indicated) showing the expression of GFP driven by the Pros promoter (green, marking EEs) and DAPI (blue). Quantification of the EEs as a percentage of total DAPI cells during aging is shown on the right. One-way ANOVA followed by Dunnett's multiple comparisons test: ****, p<0.0001. Young: n = 10, N = 3; Mid: n = 9, N = 2; Old: n = 8, N = 3. (**B**) Staining of young and old posterior midguts for YFP (green, marking ISCs), Pros (red), and DAPI (blue). Arrowheads point to Pros+ EEs (red) or Pros/YFP+ EE progenitors (green). Quantification of the proportion of EEs as a percentage of total DAPI cells as well as the proportion of EE progenitor cells as a percentage of total YFP + ISCs are shown on the right. Two-tailed t-test: ***, p=0.0005. Young: n = 9, N = 3; Old: n = 6, N = 3. n: number of posterior midguts analyzed; N: number of independent experiments performed with similar results and a similar n. Data represented as mean ± SD.

## The polycomb repressor complex is required for EE differentiation

Combined with the changes in chromatin accessibility at sequences containing motifs recognized by PcG-associated factors, these changes in H3K27 methylation suggested a change in PRCs activity or chromatin binding in aging ISCs. To test the function of PRC in ISC lineage specification directly, we asked whether loss of PRC would impact cell differentiation in the ISC lineage. We utilized the esg-Flip-Out[ts] (esg-F/O[ts]) system (*Jiang et al., 2009*) to lineage trace ISCs/EBs upon RNAi knockdown of

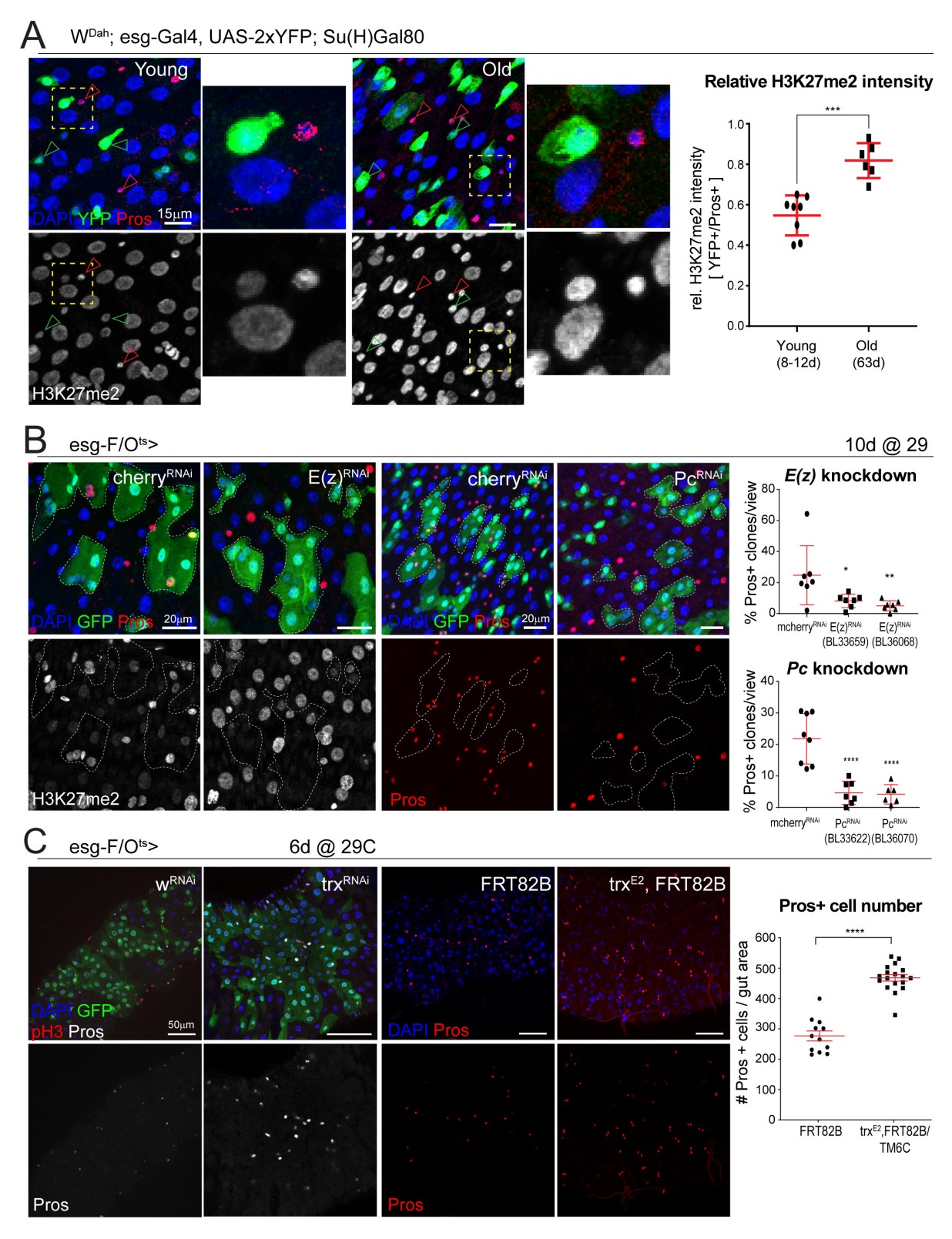

**Figure 4.** The polycomb repressor complex regulates H3K27 methylation and is required for EE cell differentiation. (A) Fluorescent immunohistochemistry on young and old posterior midguts showing ISC-specific expression of YFP (green) as well as Pros (red, marking EEs), H3K27me2 (white) and DAPI (blue). Note the higher intensity of H3K27me2 in EEs compared to other cell types in young samples and the increased H3K27me2 intensity in old ISCs (insets represent zoomed images highlighted by yellow dotted box. ISC: green arrowheads; EE: red arrowheads). The

*Figure 4 continued on next page*

*Figure 4 continued*

ratio of H3K27me2 intensity of ISCs (YFP+) over EEs (Pros+) in young and old specimens is quantified on the right. Two-tailed t-test: ***, p=0.0002. Young: n = 8, N = 3; Old: n = 6, N = 3. Differential histone post translational modification analysis of whole tissue young and old midguts is shown in *Figure 4—figure supplement 1*. (B) Clonal analysis using the esg-F/O^ts comparing control *mcherry*-RNAi to *E(z)*-RNAi clones or *mcherry*-RNAi to *Pc*-RNAi clones at 10 days of RNAi expression. Clones are marked with GFP (green) and samples were also stained for Pros (red), H3K27me2 (white) and DAPI (blue). Note the lower intensity of H3K27me2 in *E(z)*-RNAi clones. The number of clones that have at least one Pros+ cell as a percentage of all clones per view for *E(z)*-RNAi and *Pc*-RNAi are quantified on the right. One-way ANOVA followed by Dunnett's multiple comparisons: *, p=0.0265; **, p=0.0089; ****, p<0.0001. *mcherry*-RNAi: n = 7; *E(z)*-RNAi (BL33659): n = 7; *E(z)*-RNAi (BL36068): n = 7; *mcherry*-RNAi: n = 8; *Pc*-RNAi (BL33622): n = 7; *Pc*-RNAi (BL36070): n = 6, N = 1. Knockdown or either *Pc* or *E(z)* minimally affects ISC proliferation upon bacterial challenge (*Figure 4—figure supplement 2*). (C) Pros staining of esg-F/O^ts clones expressing either control (*w*-RNAi) or an RNAi against *trx* for 6 days. Animals heterozygous for the *trx^E2* mutation have more Pros+ EE cells as is shown in images of control (*FRT82B*) and trx (*trx^E2, FRT82B*) midguts stained with Pros (red). The number of Pros+ EE cells for control and *trx^E2* midguts is quantified on the right. Two-tailed t-test: ****, p<0.0001. *FRT82B*: n = 12; *trx^E2*: n = 18. n: number of posterior midguts analyzed; N: number of independent experiments performed with similar results and a similar n. Data represented as mean ± SD. Quantification of ISC proliferation after loss of *trx* is shown in *Figure 4—figure supplement 3*.

The online version of this article includes the following figure supplement(s) for figure 4:

**Figure supplement 1.** Histone post-translational modification (PTM) analysis on young and old whole midguts.
**Figure supplement 2.** Quantification of ISC proliferative response after *Ecc15* challenge upon loss of *Pc* or *E(z)*.
**Figure supplement 3.** Loss of Trithorax results in increased ISC proliferation.

the PcG methyltransferase, *E(z)*, and *Pc*. Remarkably, loss of *E(z)* or *Pc* significantly reduced the proportion of lineages containing Pros+ EE cells (*Figure 4B*). Additionally, we observed a decrease in H3K27me2 intensity upon *E(z)* knockdown (*Figure 4B*). These results implicate PRC in EE lineage specification/differentiation. *Ecc15*-induced ISC proliferation was not affected by *Pc* or *E(z)* knockdown, indicating that PRC only influences specification of ISCs and their daughter cells without impacting proliferative capacity of ISCs (*Figure 4—figure supplement 2*).

The trx group protein complexes are transcriptional activators that can counteract Pc-mediated repression (*Kassis et al., 2017*; *Schuettengruber et al., 2017*). To test whether *trx* and *Pc* act antagonistically in ISCs for EE lineage specification, we performed knockdown experiments with *trx*-RNAi. Using esg-F/O^ts, knockdown of *trx* resulted in increased numbers of Pros+ cells, as well as in larger GFP+ cell lineages, indicating that loss of *trx* promotes both ISC proliferation and EE specification (*Figure 4C*, *Figure 4—figure supplement 3*). Similarly, heterozygous *trx^E2* mutant flies exhibited increased Pros+ cell numbers as well as increased ISC proliferation (*Figure 4C*, *Figure 4—figure supplement 3A*), while knocking down *trx* in esg+ cells only also resulted in increased proliferation (*Figure 4—figure supplement 3B*). Taken together, these findings indicate that the antagonism between Pc and trx regulates EE commitment in the ISC lineage.

## Loss of *Pc* represses EE gene expression in ISCs, while loss of *trx* induces EE and cell cycle genes

To understand the mechanism by which Pc and trx regulate EE cell fate, we performed RNA-seq on either sorted ISCs (*Pc*-RNAi) or ISC/EBs (*trx*-RNAi). GSEA revealed a positive enrichment for GO terms involving ribosome biology and protein production after *Pc* knockdown (*Figure 5—figure supplement 1*). Notably, knocking down *Pc* in ISCs significantly decreased expression of several EE genes, including *ase* and *Synaptotagmin 4* (*Syt4*), and resulted in lower expression of key specification genes *scute* (*sc*) and *phyllopod* (*phyl*) (*Figure 5A,C*, *Supplementary file 5*). We also noted an increase in expression (albeit not significant) of EC genes (often encoding proteins involved in proteolysis, *Figure 5—figure supplement 2*). The misregulation of these genes is in line with our observations in *Figure 4*: EC differentiation is permitted upon *Pc* loss, but EE differentiation is not (*Figure 4B*, note the presence of large, EC-like cells in *Pc*-deficient clones). This suggests that Pc normally represses EC genes in ISCs and is required for EE gene expression.

Conversely, knockdown of *trx* resulted in a significant upregulation of many EE genes and cell cycle genes (*Figure 5B*, *Supplementary file 5*), consistent with the proposed antagonistic functions of trx and Pc on ISC lineage specification described above (*Figure 4C*, *Figure 4—figure supplement 3*). Furthermore, GSEA revealed a positive enrichment in GO terms involving neuron differentiation, plasma membrane localization, and peptide hormone secretion, all of which suggest EE cell fate (*Figure 5—figure supplement 1*).

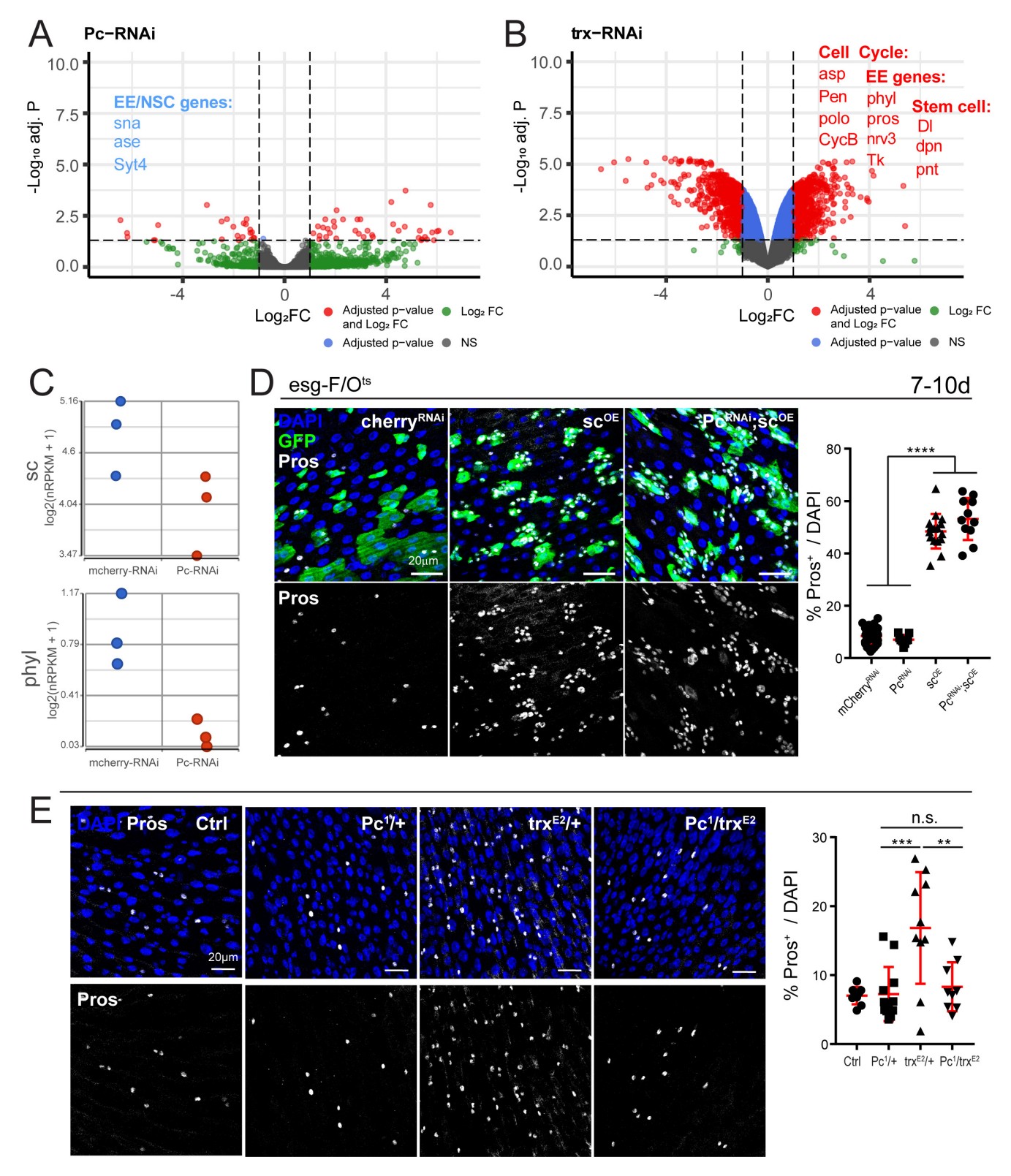

**Figure 5.** Polycomb and trithorax regulate key EE differentiation genes. (A,B) Volcano plot showing significantly differentially regulated genes after *Pc* (A) or *trx* (B) knockdown in ISCs. Genes highlighted in blue or red are significantly down- or upregulated, respectively (see also ***Supplementary file 5***). GSEA was performed on the differentially regulated genes, shown in ***Figure 5—figure supplement 1***. Promoter accessibility after *Pc* knockdown was assessed by ATAC-seq (***Figure 5—figure supplement 3***, Table 6). (C) Expression values of key EE specification genes *sc* and *phyl* decreases after *Pc-*

*Figure 5 continued on next page*

*Figure 5 continued*

RNAi expression. The increase in EC genes is depicted in *Figure 5—figure supplement 2*. (D) Overexpression of *sc* rescues EE loss after *Pc*-RNAi. Immunostaining shows esg-F/O$^{ts}$ clones (GFP+, green), Pros (white), and DAPI (blue). The proportion of EEs in total DAPI cells per view were quantified after 7d-10d of induction at 29°C. Note no changes in EE proportion after *sc* overexpression and *Pc* knockdown compared to *sc* overexpression alone. One-way ANOVA followed by Tukey's multiple comparisons test: ****, p<0.0001. *mcherry*-RNAi: n = 40; *Pc*-RNA: n = 9; *sc*-OE: n = 16; *Pc*-RNAi; *sc*-OE: n = 11; N = 2. OE: overexpression. Data represented as mean ± SD. (E) Immunostaining showing Pros+ cells in *Pc$^1$* and *trx$^{E2}$* heterozygous mutant flies and *Pc$^1$*, *trx$^{E2}$* trans-heterozygous mutant flies. Note that re-balancing *Pc* and *trx* transcript levels results in normal EE numbers. The proportion of EEs in total DAPI cells per view were quantified. One-way ANOVA followed by Tukey's multiple comparisons test: ***, p=0.0003; **, p=0.003. WT control: n = 9; *Pc$^1$*/+: n = 12; *trx$^{E2}$*/+: n = 10; *Pc$^1$*/*trx$^{E2}$*: n = 9; N = 3. Data represented as mean ± SD.

The online version of this article includes the following figure supplement(s) for figure 5:

**Figure supplement 1.** GSEA of significantly differentially regulated genes after Pc-RNAi and trx-RNAi.

**Figure supplement 2.** Expression of EC genes after *Pc* knockdown.

**Figure supplement 3.** ATAC-seq of ISCs after *Pc* knockdown.

Since key EE specification genes *ase*, *sc* and *phyl* were downregulated after *Pc*-RNAi, we asked whether we could rescue the loss of EEs by concomitant overexpression of one of these factors. To this end, we performed genetic interaction experiments in which we overexpressed wild-type, full-length *sc* (*sc*-OE) and *Pc*-RNAi together using the esg-F/O$^{ts}$ system. Expressed alone, *sc* robustly induces EE differentiation (*Figure 5D*). Notably, *sc* overexpression rescued the EE differentiation deficiency after *Pc* loss (*Figure 5D*) indicating that it acts downstream of the Pc complex in regulating EE differentiation. Thus, the presence of Pc is critical for enabling the expression of EE specification genes.

The RNA-seq data (*Figure 5A,B*) suggest that Pc and trx may be competing for common EE specification genes, as disrupting the expression of either gene tips the balance towards or against EE differentiation. We therefore wondered whether restoring the stoichiometry of both genes would restore normal EE levels. To this end, we examined heterozygous mutants *Pc$^1$* and *trx$^{E2}$* (crossed to wild-type *W$^{Dah}$*) as well as trans-heterozygous *Pc$^1$* and *trx$^{E2}$* mutants. *Pc$^1$* heterozygous mutants exhibited normal levels of EEs (*Figure 5E*), while *trx$^{E2}$* heterozygotes had higher EE levels as shown before (*Figure 4C*, *Figure 5E*). Strikingly, re-balancing *Pc* and *trx* transcript levels restored EE numbers back to wild-type proportions, thus highlighting the importance of maintaining stable stoichiometry between these factors in EE differentiation.

Since PRC is a known chromatin regulator (*Schuettengruber et al., 2017*), we asked how Pc regulates chromatin accessibility in ISCs and whether these sites are also affected in aging ISCs. To this end, we performed ATAC-seq on sorted ISCs after *Pc* knockdown. Intriguingly, though PRC is traditionally thought to maintain a repressive chromatin state, knocking down *Pc* in ISCs resulted in a moderate decrease in global chromatin accessibility (*Figure 5—figure supplement 3A*). Regions that were most differentially regulated were intergenic and intronic regions (*Figure 5—figure supplement 3B*). Differential peak analysis on promoter regions (20% of total differential peaks, *Figure 5—figure supplement 3B*) revealed a decrease in accessibility of ISC genes such as *esg* and *spdo* as well as the pro-neural gene *dpn* (*Figure 5—figure supplement 3C*), which has recently been implicated in the development of neuroendocrine tumor formation in the fly midgut (*Li et al., 2020*). Promoters with increased accessibility included homeobox transcription factors *Abd-B* and *Lim1*, as well as a *chinmo*, a BTB-zinc finger transcription factor involved in regulating stem cell self-renewal (*Dillard et al., 2018*; *Flaherty et al., 2010*). Promoters of Lim1 and chinmo were also differentially accessible in aging ISCs and their transcripts were differentially regulated (*Figure 5—figure supplement 3D*). Interestingly, EE gene promoters were not significantly affected after Pc-RNAi, suggesting their expression may be regulated independently of promoter accessibility.

## Lowering *Pc* expression inhibits age-related EE increase and gut dysplasia

We next asked whether the age-related increase in EE cell differentiation is mediated by PcG proteins in ISCs. We knocked down *Pc* in ISCs specifically using the esg-Gal4, UAS-2xEYFP/Cyo; Su(H) GBE-G80, tub-Gal80$^{ts}$/TM3 (ISC$^{ts}$) fly line and aged animals at 29°C for 28 days. Strikingly, the progressive increase in the proportion of Pros+ EE cells and of esg+/Pros+ EE progenitor cells observed in aging control animals was significantly reduced after *Pc* knock-down (*Figure 6A*). The decrease in

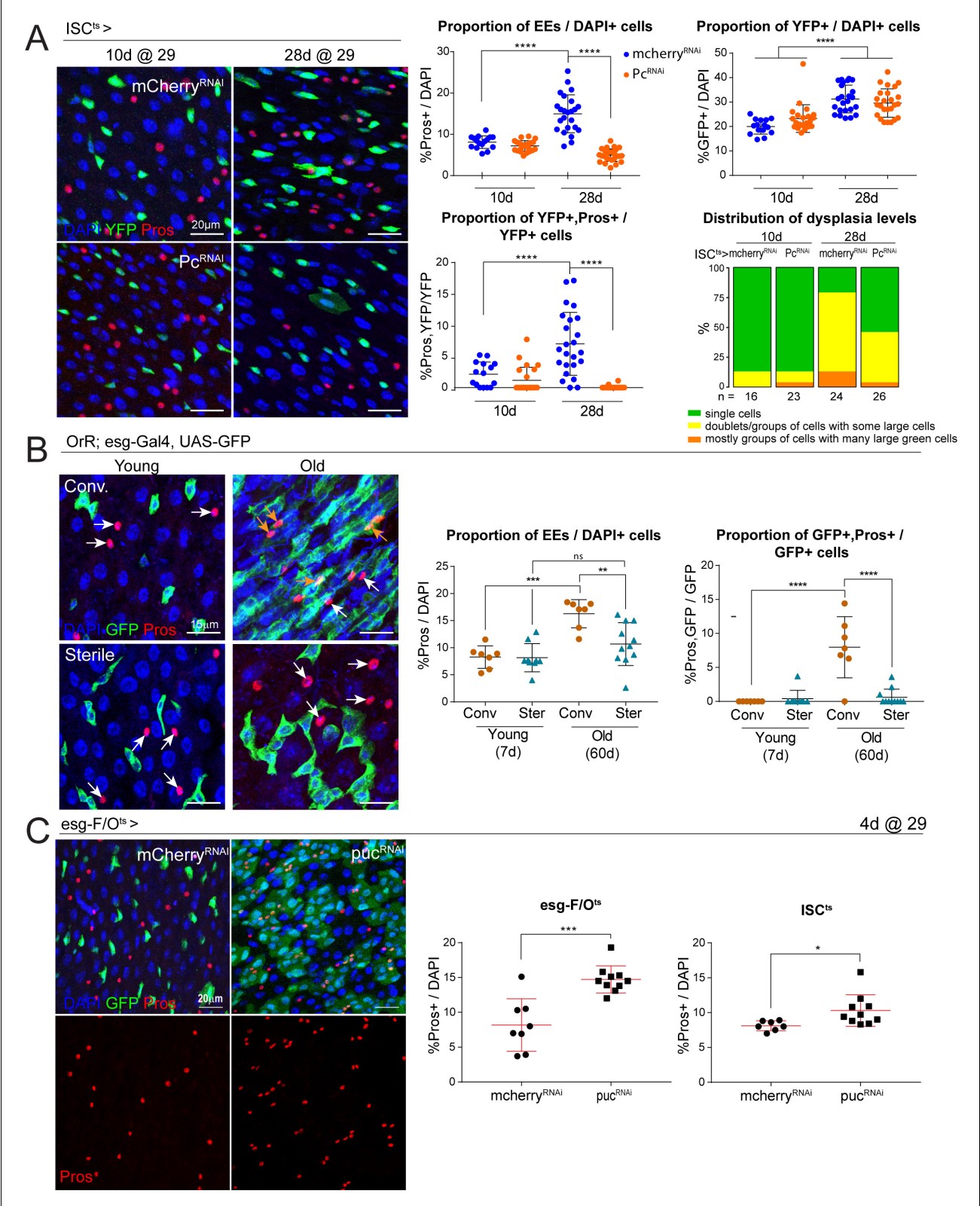

**Figure 6.** Long-term downregulation of *Polycomb* levels inhibits EE increase, which is elevated after stress signaling. (**A**) Midguts from flies aged at 29° C for short- (10d) and long-term (28d) knockdown of *Pc* specifically in ISCs using the ISC[ts] fly line. Immunostaining shows ISCs (YFP+, green), EEs (Pros +, red) and DAPI (blue). The proportion of EE cell numbers of total DAPI cells per view, as well as the proportion of EE progenitors of total ISCs per view, are quantified on the right. One-way ANOVA followed by Tukey's multiple comparisons test: ****, p<0.0001. Distribution of dysplasia levels across

*Figure 6 continued on next page*

*Figure 6 continued*

short- and long-term knockdown of *Pc* in ISCs compared to control cherry knockdown. *mcherry*-RNAi (10d): n = 16; *Pc*-RNAi (10d): n = 23; *mcherry*-RNAi (28d): n = 24; *Pc*-RNAi (28d): n = 26; N = 1. (B) Comparison of the number of EEs in conventionally versus axenically (sterile) raised flies. Staining shows ISCs/EBs marked by GFP (green), EEs marked by Pros (red) and DAPI (blue). The proportion of EEs in total DAPI cells is quantified on the right as well as the proportion of EE progenitors in the total ISC population. Levels of dysplasia correlated with numbers of EE progenitors regardless of age (*Figure 6—figure supplement 1*). One-way ANOVA followed by Tukey's multiple comparisons test: **, p=0.0033; ***, p=0.0002; ****, p<0.0001. Young, conv.: n = 7; young, sterile: n = 9; old, conv.: n = 7; old, sterile: n = 11; N = 1. (C) Immunostaining of posterior midguts after overexpression of *mcherry*-RNAi or *puc*-RNAi for 4 days using esg-F/O$^{ts}$. Quantification of EE cells as a proportion of total DAPI cells per view using esg-F/O$^{ts}$ or ISC$^{ts}$ to drive expression is shown on the right. Two-tailed t-test: *, p=0.0268; ***, p=0.0002. esg-F/O$^{ts}$>*mcherry*-RNAi: n = 8; *puc*-RNAi: n = 10; N = 2. ISC$^{ts}$>*mcherry*-RNAi: n = 7; *puc*-RNAi: n = 10; N = 1. n: number of posterior midguts analyzed; N: number of independent experiments performed with similar results and a similar n. Data represented as mean ± SD.

The online version of this article includes the following figure supplement(s) for figure 6:

**Figure supplement 1.** The proportion of enteroendocrine cells correlates with dysplasia levels.

Pros+ cells was not due to a loss of ISCs, as ISC numbers were not changed at either the young or old time-points, suggesting Pc is mainly required for EE specification, but does not affect ISC viability (*Figure 6A*). Importantly, long-term *Pc* knock-down also reduced levels of dysplasia in aged animals (*Figure 6A*), suggesting that the increase in EE cells contributes to age-related intestinal dysplasia. Overall, these results identify an important role for PRC in EE lineage commitment of ISCs, and confirm that changes in Pc activity contribute to the age-related skewing of ISC lineage fidelity. Moreover, these findings implicate higher EE levels in disrupting intestinal homeostasis during aging.

## Overactivation of JNK promotes EE lineage

Loss of epithelial homeostasis in the aging *Drosophila* intestine is associated with increased bacterial load and commensal dysbiosis, which result in the chronic elevation of inflammatory signaling and activation of ISC proliferation, causing dysplasia (*Biteau et al., 2008*; *Buchon et al., 2009*; *Guo et al., 2014*; *Jasper, 2020*; *Ren et al., 2007*). Interestingly, the increase in EE cell numbers was robustly and consistently observed in animals that exhibited age-related epithelial dysplasia. In old animals that did not exhibit dysplasia, however, the increase in EE cells was less pronounced. In fact, dysplasia levels (categorized based on GFP+ cell number, distribution and morphology; *Biteau et al., 2010*) correlated with the percentage of EE progenitors regardless of age (*Figure 6—figure supplement 1*). This suggested that the shift in ISC lineage fidelity may be a consequence of commensal dysbiosis and increased inflammation. To test this idea, we raised and aged flies under conventional or axenic conditions and quantified the proportion of EE cells in young and old animals (*Figure 6B*). Indeed, while the proportion of EEs and EE progenitors increased in conventionally aged flies, these increases were significantly reduced in axenically aged flies (*Figure 6B*).

A major stress signaling pathway activated in ISCs responding to elevated inflammation is the Jun-N-terminal kinase (JNK) pathway, which has also been implicated in modulating Pc complex activity (*Lee et al., 2005*; *Roumengous et al., 2017*). We therefore asked whether JNK activation may also influence EE fate in the ISC lineage. Knocking down the negative regulator of JNK, *puckered* (*puc*), in the ISC lineage (using the esg-F/O$^{ts}$ system) or in ISCs specifically (ISC$^{ts}$), resulted in a significant increase in the proportion of Pros+ EE cells after 4 days of over-expression (*Figure 6C*). These results support a role for age-related inflammation and stress signaling, particularly activation of the JNK pathway in ISCs, in influencing ISC lineage choice towards the EE fate.

## Discussion

Our study identifies a loss of lineage fidelity in ISCs of aging flies that results in an imbalance of EE vs EC differentiation and contributes to epithelial dysplasia. This loss of lineage fidelity is caused by age-related deregulation of Pc function in ISCs, resulting in de-repression and preferential expression of EE genes. We propose that the chronic activation of stress signaling in ISCs, triggered by local and systemic inflammatory stimuli in the aging intestine, promotes the deregulation of Pc-controlled gene activity (*Figure 7*). This is supported by the fact that genetically elevating JNK activity in ISCs disrupts lineage fidelity and causes an increase in the proportion of EEs in the gut epithelium.

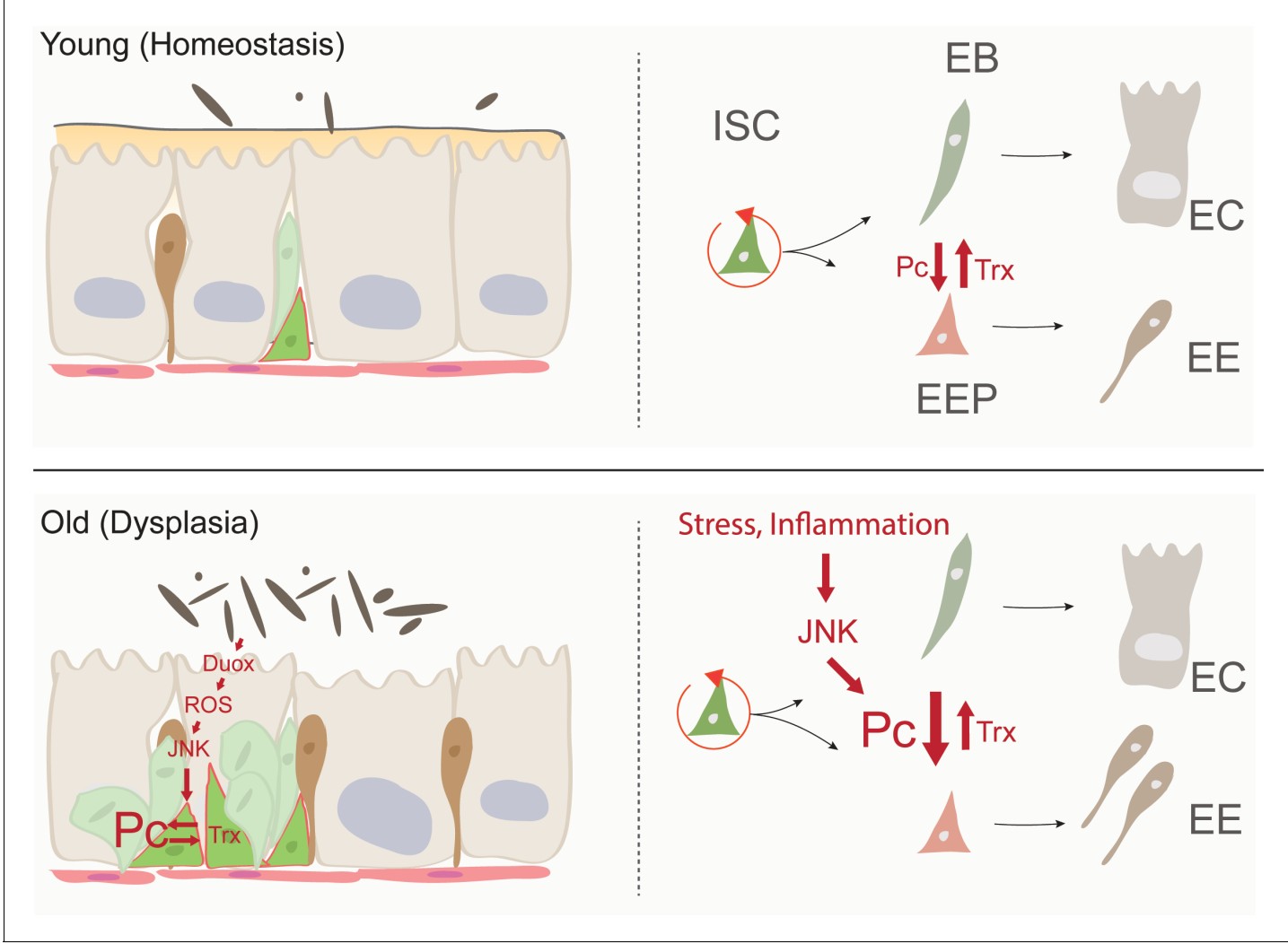

**Figure 7.** Model of age-related deregulation of ISC lineage fidelity by perturbed Polycomb and Trithorax balance. In young flies, balanced Pc and trx activity maintain appropriate specification of ISCs into the EE or EC lineage. In old flies, elevated JNK activity in ISCs (a consequence of commensal dysbiosis – induced inflammation and ROS production) in ISCs perturbs this balance, resulting in elevated commitment into the EE lineage.

The increase in EE numbers contributes to epithelial dysplasia in the aging gut, as EEs can promote ISC proliferation (*Amcheslavsky et al., 2014*). The stress-induced changes in lineage fidelity in ISCs thus likely set up a vicious cycle that causes progressive dysplasia and results in disruption of epithelial structure and function in the aging intestine.

Our scRNA-seq analysis of the *Drosophila* gut is consistent with a recent scRNA-seq study from young flies (*Hung et al., 2020*), but also captures the changes in intestinal cell states across aging. Notably, we observe the well-characterized age-associated increase in mitotically active ISCs, but also identify a unique 'stressed ISC' cell population that increases with age. The transcriptional signature distinguishing this cell population encompasses over 25% of significantly upregulated genes observed in our bulk RNA-seq study from purified old ISCs, supporting the robustness and complementarity of the two methods. This transcriptional signature was enriched in genes involved in glutathione metabolic processes, chaperone-mediated protein folding, response to heat, and regulation of cytoskeleton organization, consistent with a stressed or damaged cell state. The appearance of this 'stressed' stem cell population is further consistent with the previously described increase in inflammatory and oxidative stress in the aging intestinal epithelium, and may reflect the age-associated accumulation of oxidative, proteostatic and genomic damage in these cells (*Buchon et al., 2009*; *Guo et al., 2014*; *Hochmuth et al., 2011*; *Park et al., 2012*; *Rodriguez-Fernandez et al.,*

*2019*; *Siudeja et al., 2015*; *Sousa-Victor et al., 2017*). Overall, the majority of old ISCs reside in an activated cell state (~50%), whereas the 'stressed ISC' population makes up only a small percentage (<10%). How this 'stressed' population affects tissue homeostasis requires further studies. Intriguingly, the p53 and DNA repair pathways are upregulated in this cell population, while cell cycle genes are repressed, indicating that these cells may represent a correlate to mammalian senescent cells.

The EE progenitor population within Cluster five not only increases in size with age, but also upregulates pro-neural genes that are markers for neural stem cells (NSCs). It was recently shown that in a neuroendocrine tumor model, ISCs undergo an identity switch that results in the acquisition of NSC-like features (*Li et al., 2020*). This NSC gene signature was also upregulated in old ISCs analyzed by bulk RNAseq, suggesting that a general upregulation of these genes may contribute to age-related ISC phenotypes.

Our ATAC-seq data from purified ISC populations revealed only moderate changes in chromatin organization in ISCs of increasingly older animals, suggesting that ISC gene regulation is tightly controlled throughout life. At the same time, the significant increase in H3K27 dimethylation levels in aging ISCs, the fact that H3K27me2 levels are higher in EEs than in ISCs in young guts, and the observation that E(z)-mediated methylation of H3K27 is required for EE specification, all support a role for increases in H3K27me2 in skewing ISC identity towards the EE fate. Despite the high genomic abundance of H3K27me2, which accounts for up to 70% of total histone H3 (*Ebert et al., 2004*; *Ferrari et al., 2014*), the functional role of H3K27me2 remains largely uncharacterized. The broad genomic distribution of H3K27me2 was shown to suppress aberrant gene activation by controlling enhancer fidelity in mammals (*Ferrari et al., 2014*), and access to transcription factors and RNA Pol II to DNA in flies (*Lee et al., 2015*). If and how genomic abundance and/or distribution of H3K27me2 affects cell identity or other cellular functions has not been well explored. One study found that perturbing the ratio of H3K27me2/H3K27me3 in mouse embryonic stem cells (ESCs) affected the acquisition and repression of specific fates of these cells (*Juan et al., 2016*), indicating the importance of appropriate regulation of these marks in different cell types. The differential abundance of H3K27me2 that we observed in young ISCs and EEs further supports the importance of dynamic H3K27me2 regulation in the ISC lineage and of appropriate control of this mark to maintain lineage commitment. The loss of EE cell differentiation upon *Pc* or *E(z)* depletion in ISCs further supports a critical role for PRC in regulating H3K27 methylation status and thereby lineage fidelity. Of note, the expression of *Pc*, *E(z)* and other PcG genes, as well as the expression of *trx*, *Trl* and *z*, was not significantly altered in aging ISCs, suggesting that aging most likely affects their post-transcriptional regulation and/or function.

Age-related changes in H3K27 methylation have been reported in mammalian SC populations: in aging HSCs and muscle SCs, the H3K27me3 signal exhibits broader coverage and increased intensity at transcriptional start sites and intergenic regions (*Liu et al., 2013*; *Sun et al., 2014*), indicating that there is an evolutionarily conserved effect of aging on PRC function in tissue stem cells. It would be intriguing to explore whether these alterations in H3K27me3 may underlie the age-related dysfunction in lineage potential observed in HSCs of old mice (*Elias et al., 2017*; *Sun et al., 2014*).

Since loss of *Pc* induces the expression of EC genes, represses EE gene expression, and results in less accessible chromatin associated with ISC identity genes (*esg*, *spdo*) as well as pro-neural genes (*dpn*), we propose that Pc activity regulates multiple aspects of ISC specification. Despite the upregulation of EC genes after Pc depletion, ISCs did not spontaneously differentiate, ISC numbers remained normal and ISCs could still mount a proliferative response to infection. Thus, ISC function remained largely intact suggesting the primary function of Pc in ISCs is to regulate lineage commitment.

The contrasting function of trx in EE differentiation is consistent with the known antagonism between trx and Pc complexes (*Schuettengruber et al., 2017*) and exemplifies a tightly regulated interplay of these systems in lineage commitment. In addition to upregulating EE genes, loss of trx also induced cell cycle genes and ISC proliferation, suggesting additional roles in controlling ISC function. This finding is in line with a recently published study showing TrxG factors Kismet and Trr limit ISC proliferation in the fly midgut (*Gervais et al., 2019*). The fact that this group did not report changes in the EE lineage most likely reflects functional differences across TrxG complexes, the composition of which varies greatly (*Geisler and Paro, 2015*).

A similar function in stem cell specification has been described for PRC1 and PRC2 in the mouse, where both complexes are important to preserve ISC and progenitor cell identity in the gut, while regulating specification into specific daughter cell lineages (*Chiacchiera et al., 2016a*; *Chiacchiera et al., 2016b*; *Koppens et al., 2016*). Loss PRC1 function in the intestinal epithelium resulted in impairment of ISC self-renewal via de-repression non-intestinal lineage genes as well as negative regulators of the Wnt signaling pathway (*Chiacchiera et al., 2016a*). Interestingly, the effects of PRC1 loss were independent of H3K27me3, revealing instead the role of H2AK119 mono-ubiquitination (*Chiacchiera et al., 2016a*). PRC2 was shown to be important in ISCs only during damage-induced regeneration (*Chiacchiera et al., 2016b*). In contrast, another study found significant degeneration of the SC compartment under homeostatic conditions as well (*Koppens et al., 2016*). Reconciling these findings, a recent study revealed that in the absence of PRC2, mammalian cells shed H3K27me3 exclusively by replicational dilution of modified nucleosomes, and that the effects of PRC2 deletion are thus only observed in lineage progeny rather than in stem cells themselves (*Jadhav et al., 2020*). Both previous studies are in agreement, however, that PRC2 controls cell fate decisions, as loss of PRC2 leads to an accumulation of secretory cells, evidently due to de-repression of the secretory lineage master regulator, *Atoh1*, resulting in ISC differentiation (*Chiacchiera et al., 2016b*; *Koppens et al., 2016*). It remains unknown how aging affects PRC function and H3K27 methylation in the mammalian intestine.

Only a few studies have rigorously investigated age-associated changes in the mammalian intestine on a histological and cell-type-specific level. Nalapareddy et al. reported changes in crypt architecture, decreased mitotic potential of ISCs and an increase in the secretory cell lineage, most likely due to increased Atoh1 expression (*Nalapareddy et al., 2017*). Given the role of PRC2 in regulating the secretory lineage in both the mammalian intestine and the fly, it is tempting to speculate a conserved age-related increase in the secretory lineage that stems from deregulation of PRC. Our results support a role for increased stress signaling in driving this lineage imbalance, as overactive JNK in ISCs promotes EE differentiation. While JNK signaling has been reported to suppress Pc complex function (*Lee et al., 2005*), our data indicate that in the ISC lineage, this interaction is more complex, as both JNK activation promotes EE specification, while Pc knockdown inhibits EE specification. Further studies are needed to explore the molecular mechanisms mediating JNK/Pc interactions in the ISC lineage.

Chronic elevation of inflammatory signaling is a well-characterized hallmark of the aging fly intestine (*Biteau et al., 2008*; *Buchon et al., 2009*; *Li et al., 2016*) and a hallmark of many intestinal disorders including inflammatory bowel disease (IBD), infection and colorectal cancer. Alterations in EE cell numbers and secretory activity have been reported to play a role in many diseases (*Worthington et al., 2018*). In IBD, for example, EE cells were shown to contribute to pathogenesis by producing pro-inflammatory cytokines (*Friedrich et al., 2015*). In another study, increased numbers of EE cells were reported in human patients with chronic ulcerative colitis, potentially promoting IBD associated neoplasias (*Gledhill et al., 1986*). Notably, we show that lowering EE numbers by long-term depletion of Pc in ISCs inhibited age-induced intestinal dysplasia, supporting a pathological role for EEs in aging.

The role of epigenetic alterations, and specifically the role of PRC, in inflammatory diseases and cancer is still under investigation (*Comet et al., 2016*; *Ventham et al., 2013*). It was recently reported that suppressing EZH2 activity ameliorates experimental intestinal inflammation and delays the onset of colitis-associated cancer. However, these effects may be a consequence of EZH2 suppression in myeloid cells rather than in intestinal stem cells (*Zhou et al., 2019*). Disruption in PRC2 function may also underlie human cancers, where PRC2 is often hyperactive or overexpressed (*Comet et al., 2016*). Activating EZH2 mutations, which increase total H3K27me3 levels (*Sneeringer et al., 2010*; *Yap et al., 2011*), increase tumor survival and growth in pre-clinical models (*Béguelin et al., 2013*) and are found in up to 24% of diffuse large B cell lymphomas (*Morin et al., 2010*). Additional work will be needed to establish whether age-related changes in PRC activity contribute to the increased onset of gastrointestinal cancers during aging.

Taken together, our findings provide evidence for altered ISC cell states in old flies that affect intestinal homeostasis and contribute to tissue dysplasia. Our results exemplify the importance of maintaining appropriate lineage decisions, as overproduction of EE cells is detrimental to the epithelium, but can be rescued by re-balancing the system towards normal EE numbers. We propose age-associated deregulation of lineage fidelity of ISCs due to elevated stress and misregulation of Pc as

key drivers of functional decline of the intestinal epithelium. Pc group proteins may thus represent valuable therapeutic targets for age-related morbidities.

# Materials and methods

## Key resources table

| Reagent type (species) or resource | Designation | Source or reference | Identifiers | Additional information |
|---|---|---|---|---|
| Genetic reagent (*D. melanogaster*) | UAS-mcherry-RNAi | Bloomington *Drosophila* Stock Center | BDSC:35785; RRID:BDSC 35785 | |
| Genetic reagent (*D. melanogaster*) | UAS-w-RNAi | Bloomington *Drosophila* Stock Center | BDSC:25785; RRID:BDSC 25785 | |
| Genetic reagent (*D. melanogaster*) | UAS-Pc-RNAi | Bloomington *Drosophila* Stock Center | BDSC:36070; RRID:BDSC 36070 | |
| Genetic reagent (*D. melanogaster*) | UAS-Pc-RNAi | Bloomington *Drosophila* Stock Center | BDSC:33622; RRID:BDSC 33622 | |
| Genetic reagent (*D. melanogaster*) | UAS-E(z)-RNAi | Bloomington *Drosophila* Stock Center | BDSC:33659; RRID:BDSC 33659 | |
| Genetic reagent (*D. melanogaster*) | UAS-E(z)-RNAi | Bloomington *Drosophila* Stock Center | BDSC:36068; RRID:BDSC 36068 | |
| Genetic reagent (*D. melanogaster*) | trx$^{E2}$, FRT82B/Tm6C | Bloomington *Drosophila* Stock Center | BDSC:24160; RRID:BDSC 24160 | |
| Genetic reagent (*D. melanogaster*) | FRT82B | Bloomington *Drosophila* Stock Center | BDSC:2051; RRID:BDSC 2051 | |
| Genetic reagent (*D. melanogaster*) | Pc$^1$/TM1 | Bloomington *Drosophila* Stock Center | BDSC:1728; RRID:BDSC 1728 | |
| Genetic reagent (*D. melanogaster*) | UAS-eGFP | Bloomington *Drosophila* Stock Center | BDSC:6874; RRID:BDSC 6874 | |
| Genetic reagent (*D. melanogaster*) | OregonR | Bloomington *Drosophila* Stock Center | BDSC:5; RRID:BDSC 5 | |
| Genetic reagent (*D. melanogaster*) | UAS-trx-RNAi | Vienna *Drosophila* Resource Center | Transformant ID: GD37715 | |
| Genetic reagent (*D. melanogaster*) | UAS-puc-RNAi | Vienna *Drosophila* Resource Center | Transformant ID: GD3018 | |
| Genetic reagent (*D. melanogaster*) | UAS-sc-3HA | Fly ORF | F000085 | |
| Genetic reagent (*D. melanogaster*) | Pros$^{V1}$-Gal4 | *Grosjean et al., 2001* | N/A | J. Korzelius |
| Genetic reagent (*D. melanogaster*) | W$^{Dah}$ | L. Partridge | N/A | |
| Genetic reagent (*D. melanogaster*) | esg-Gal4, UAS-GFP | S. Hayashi, RIKEN, Japan | N/A | |
| Genetic reagent (*D. melanogaster*) | esg-Gal4, UAS-2xEYFP/Cyo; Su(H)GBE-G80/TM3 | *Wang et al., 2014* | N/A | S. Hou, NIH, USA |
| Genetic reagent (*D. melanogaster*) | esg-Gal4, UAS-2xEYFP/Cyo; Su(H)GBE-G80, tub-Gal80$^{ts}$/TM3 (ISC$^{ts}$) | *Wang et al., 2014* | N/A | S. Hou, NIH, USA |

*Continued on next page*

*Continued*

| Reagent type (species) or resource | Designation | Source or reference | Identifiers | Additional information |
|---|---|---|---|---|
| Genetic reagent (*D. melanogaster*) | w; esg-Gal4, UAS-GFP, tubGal80ts/Cyo; UAS-Flp, act>>CD2>>G4/TM6B (esg-F/O$^{ts}$) | B. A. Edgar | N/A | |
| Antibody | anti-Prospero (mouse monoclonal) | DSHB | Cat# MR1A; RRID:AB_528440 | IF (1:250) |
| Antibody | anti-H3K27me2 (rabbit polyclonal) | Abcam | Cat# ab24684; RRID:AB_448222 | IF (1:500) |
| Antibody | anti-GFP (chicken polyclonal) | Abcam | Cat# ab13970; RRID:AB_300798 | IF (1:2000) |
| Antibody | anti-phospho-histone H3 (rabbit polyclonal) | Sigma-Aldrich | Cat# 06–570 RRID:AB_310177 | IF (1:1000) |
| Other | Hoechst 33342 | Thermo Fisher Scientific | Cat# 62249 | IF (1:1000) |
| Commercial assay or kit | RNeasy Mikro Kit | Qiagen | Cat# 74004 | |
| Commercial assay or kit | Nextera DNA Sample prep kit | Illumina | Cat# FC-131–1024 | |
| Commercial assay or kit | MiniElute Reaction Cleanup kit | Qiagen | Cat# 28204 | |
| Commercial assay or kit | NEBNext High-Fidelity 2X PCR Master Mix | New England Biolabs | Cat #M0541 | |
| Commercial assay or kit | Chromium Single Cell 3′ Library and Gel bead kit v2 | 10x Genomics | PN-120237 | |
| Software, algorithm | FIJI is just ImageJ (FIJI) | Schindelin et al., 2012 | RRID:SCR_002285 | https://fiji.sc/ |
| Software, algorithm | GraphPad Prism | GraphPad | RRID:SCR_002798 | https://www.graphpad.com/scientific-software/prism/ |
| Software, algorithm | Partek Flow Genomic Analysis Software | Partek | | https://www.partek.com/partek-flow/ |
| Software, algorithm | Adobe Photoshop and Illustrator | Adobe | RRID:SCR_014199; RRID:SCR_010279 | https://www.adobe.com/ |

## *Drosophila* husbandry

Flies were raised on standard yeast/molasses food. Unless otherwise indicated, flies were maintained and aged at 25 °C with 65% humidity on a 12 hr light/dark cycle. Mated (for at least 3 days) females were used in all experiments.

## Axenic fly cultures

Flies were raised axenically according to the protocol described in *Guo et al., 2014*. In short, eggs were collected for 16 hr, dechorionated for 3 min in 2.7% sodium hypochlorite (2-fold diluted bleach) and washed twice with sterile, distilled water. Embryos were then transferred into sterile food vials in a biosafety cabinet and covered with 70% sterile glycerol. Adult flies were kept in a biosafety cabinet and transferred to new sterile food every 2–3 days. To confirm axenia, gut homogenates were plated onto Nutrient agar (Difco) plates and incubated at 29 °C for 3 days.

## Challenge with Ecc15

*Ecc15* was cultured in LB medium for 16 hr at 30°C. The culture was pelleted (15 ml/vial), re-suspended in residual LB medium and diluted 1:1 in 5% sucrose in water. Mock and infection vials were prepared by placing a pre-cut Whatman Filter Paper disk onto the top of the fly food and then adding 150 µl of either 2.5% sucrose in water (mock) or the prepared *Ecc15* solution (infection). Flies were starved in empty vials 2–3 hr prior to treatment and were treated for 16 hr at which point they were dissected.

## Conditional expression of UAS-linked transgenes

We used the TARGET system (*McGuire et al., 2004*) to conditionally express UAS-linked transgenes in ISCs using the ISC[ts] fly line, allowing for ISC-specific expression. ISC[ts] virgin females were crossed to males carrying a UAS-linked RNAi construct and these crosses and progeny were kept at 18°C (permissive temperature) to restrict transgene expression. Eclosed progeny was collected over a 2- to 3-day timespan and mated females were shifted to 29°C, the restrictive temperature that allows for transgene expression, 5–7 days after eclosion. Flies were kept at 29°C for indicated lengths of time after which point they were dissected.

For lineage-tracing experiments, we used the Flp-Out method (*Jiang et al., 2009*; *Theodosiou and Xu, 1998*) using the temperature-sensitive esg-F/O[ts] fly line. Here, similar to TAR-GET experiments, flies were crossed and progeny were kept at 18°C. Transgene expression and lineage tracing was induced in young (5–7 day-old) flies by transferring flies to 29°C and they were maintained there for indicated lengths of time for each experiment.

## ISC isolation, ATAC-seq and RNA-seq

### ATAC-seq

Two cohorts of aging flies were used to perform ATAC-seq analyses, herein referred to as Cohort 1 and Cohort 2. Cohort 1 was used for further in-depth analyses. For these experiments, whole midguts were dissociated into single cells and ISCs were sorted by Fluorescent Activated Cell Sorting (FACS) as we have previously described (*Tauc et al., 2014*). In short, midguts from flies of $W^{Dah}$; *esg-Gal4, UAS-2xEYFP; Su(H)GBE-Gal80,* were dissected in 1xPBS, 1% Bovine Serum Albumin (BSA) and dissociated in 0.5% Trypsin-EDTA solution for less than 2 hr at room temperature (RT), during which dissociated cells were collected periodically every 20–30 min, re-suspended in 1xPBS, 1%BSA and 2%FBS and kept on ice until sorting. A BD Biosciences FACSAria II flow cytometer cell sorter was used for cell sorting. For ATAC-seq analysis, the protocol described in *Buenrostro et al., 2015* was used with minor adjustments. For each ATAC-seq sample, 100,000 YFP+ ISCs were sorted by FACS and cells were immediately spun down at 4°C, washed, and then lysed and released nuclei were used in a transposition reaction using Nextera DNA Sample prep kit according to manufacturer's protocol (Illumina). Transposed DNA was purified using MiniElute Reaction Cleanup kit (Qiagen) and eluted in 10 µl water. Purified transposed samples were amplified using NEBNext High-Fidelity 2X PCR Master Mix (New England Biolabs) per manufacturer's user guide. Generated libraries were quantified by qPCR to determine the cycle number at which each sample produced 25% of maximum fluorescent intensity. The fluorescent intensity was used to estimate the number of additional PCR cycles required for each sample. Libraries were purified using SPRIselect beads (Beckman Coulter) to remove contaminating primer dimers. Library quality was assessed with dsDNA HS Assay kit (Thermo Fisher) and quantitated by Qubit 3.0 Fluorometer (Thermo Fisher). Pooled libraries were quantitated by Kapa Library Quantification Kit (Kapa Biosystems) and sequenced on Illumina HiSeq2500 to generate 30 M of pair end 50 bp reads per library.

### RNA-seq

For RNA-seq, between 62,000–210,000 ISCs were sorted directly into Buffer RLT and RNA was isolated using the RNeasy Mikro Kit (Qiagen). QC of samples was done to determine RNA quantity and quality prior to the processing by low input RNA-seq method. The concentration of RNA samples was measured using DS-11 spectrophotometer (DeNovix) and the integrity of RNA was determined by 2100 Bioanalyzer (Agilent Technologies). Approximately few ng of total RNA was used as an input material for the library generation using SMART-seq v4 Ultra Low Input RNA kit (Clontech). Size of the libraries was confirmed using 4200 TapeStation and High Sensitivity D1K screen tape (Agilent Technologies) and their concentration was determined by qPCR based method using Library quantification kit (KAPA). The libraries were multiplexed and then sequenced on Illumina HiSeq4000 (Illumina) to generate >30M of single end 50 base pair reads.

### RNA-seq and ATAC-seq after polycomb or trithorax knockdown

For *Pc*-RNAi experiments, virgins of the ISC[ts] fly line were crossed with m*cherry*-RNAi or *Pc*-RNAi (BL36070) males and maintained at 18°C. Progeny was collected and shifted to 29°C 4–7 days after eclosion. Flies were kept at 29°C for 8–11 days at which point they were dissected and ISCs were

isolated for ATAC-seq and RNA-seq as described above. Three cohorts [n = 3] containing ISCs from ~120 midguts were used for each condition. For RNA-seq after *trx* knockdown, the esg^ts was used to drive expression of *w*-RNAi or *trx*-RNAi (VDRC: GD37715). Progeny was shifted to 29°C for 7 days at which point flies were dissected and esg+ cells were isolated for RNA-seq as described above. Two cohorts were used for each condition [n = 2].

## Single-cell RNA-seq

For the scRNA-seq experiments, we used the wild-type W^Dah fly line and dissected whole midguts at young (~7 days), mid-age (~30 days) and old time-points (~60 days). In total, we had two young, one mid-age and two old samples. Whole midguts were dissected in 1xPBS, 1%BSA and dissociated at RT using 0.5% Trypsin-EDTA solution. Midguts were agitated by vortexing every 30 min and dissociated cells were removed from the enzyme solution and placed on ice in 1xPBS, 1%BSA and 2%FBS every 30 min. Once the tissue was completely dissociated, cells were FACS sorted to ensure single cells were obtained after which cell viability was measured using a Vi-CELL analyzer to confirm >80% viability. The cells were then immediately applied to the 10XGenomics chromium platform.

Sample processing for single-cell RNA-seq was done using Chromium Single Cell 3' Library and Gel bead kit v2 following manufacturer's user guide (10x Genomics). The cell density and viability of single-cell suspension were determined by Vi-CELL XR cell counter (Beckman Coulter). All the processed samples had very high percentage of viable cells (>90%). The cell density was used to impute the volume of single cell suspension needed in the reverse transcription (RT) master mix, aiming to achieve ~6000 cells per sample. cDNAs and libraries were prepared following manufacturer's user guide (10x Genomics). Libraries were profiled on 2100 Bioanalyzer (Agilent Technologies) using High Sensitivity DNA kit (Agilent Technologies) and quantified using Kapa Library Quantification Kit (Kapa Biosystems). Each library was sequenced in one lane of HiSeq4000 (Illumina) following manufacturer's sequencing specification (10x Genomics).

## Bioinformatics analyses
### ATAC-seq analysis (Cohort 1)

ATAC-seq reads were aligned to the *Drosophila melanogaster* reference genome (BDGP6) using HTSeqGenie (version 4.12.0). Briefly, reads shorter than 18 bp were filtered out and GSNAP (version 2013-11-102) was used to map the reads to the BDGP6 reference genome. Duplicated reads have been were marked with Picard (2.21.9) and bam files belonging to the same group were merged using bamtools (2.3.0). MACS2 (2.1.0) with parameters (–`p-value 0.01`, –`shift −100`, –`extsize 199`) was used to call the peaks. Highly reproducible peaks from replicates were assigned by the Irreproducibility Discovery Rate (IDR) ENCODE framework. Reads within overlapping peaks were counted by using the summarizeOverlaps method from the GenomicAlignments (1.18.1) R package. Counts were normalized using the TMM method, and then transformed using the voom method. Differential peaks were identified by limma (version 3.38.3) R package. We used the Benjamini-Hochberg method for adjusting p-values for multiple testing. Peaks with an adjusted p-value<0.05 in the differential accessibility test were considered significantly differential. PCA plot was performed on normalized counts (cpm) using PCAtools (1.2.0) R package. Volcano plots were generated using EnhancedVolcano (1.4.0) R package. Peak coverage heatmaps were generated by deepTools (2.0). Homer (4.8) was used to find enriched motifs in differentially accessible promoter peaks. Specific fly motifs were obtained from JASPAR 2020 collection (http://jaspar.genereg.net/) and from DMMPMM database (http://autosome.ru/DMMPMM/). Data were visualized with ggplot2 (3.2.1) R package.

### ATAC-seq analysis (Cohort 2)

First, low-quality bases were removed from the beginning and the end of reads followed by trimming adapter sequencing, and filtering out sequences that were shorter than 35 bases using Trimmomatic (*Bolger et al., 2014*). Reads originating from mitochondria DNA were removed by aligning all reads to the dm6 reference genome using bowtie2 (*Langmead and Salzberg, 2012*), identifying their mapping location using SAMtools 1.7 (*1000 Genome Project Data Processing Subgroup et al., 2009*), and filtering out those reads that aligned to mDNA using an in-house Python script. At the end of the preprocessing steps, the number of pair-end reads varied between 11M to 15M per sample.

To visualize genomic coverage of the sequencing alignment, BEDTools' filter utility (*Quinlan and Hall, 2010*) was used to remove read alignments that were suspected of being an artifact. Then, peaks located in listed black genomic region were removed by running BEDTools' intersect utility. Problematic genomic regions were downloaded from https://github.com/Boyle-Lab/Blacklist/tree/master/lists. The remaining alignments of all samples were merged from the same age group by running SAMtools' merge utility (*1000 Genome Project Data Processing Subgroup et al., 2009*). In order to create a coverage file (bigwig) normalized by RPGC, for each age group, deepTools bamcoverage utility (*Ramírez et al., 2016*) was used. deepTools multiBigwigSummary utility was used to assemble all coverage info into one 'ntz' file, which then was used as an input to plot correlations and PCA graphs using deepTools plotCorrelation and plotPCA, respectively.

For the purpose of locating open genomic areas, an ENCODE recommended pipeline, developed by the Kundaje lab at Stanford University was used (available on Github at https://github.com/ENCODE-DCC/atac-seq-pipeline). The pipeline performs end to end quality control and analysis of ATAC-seq data. The pipeline was run three times, each time on the replicates pre-processed fastq files of one age group, young, mid, or old. The pipeline first aligned reads to the dm6 reference genome using bowtie2. Total alignment rates were 87.99%, 90.70% and 86.46% for young samples, 77.40% and 73.16% for mid age samples, and 96.55%, 82.29% and 80.00% for old age samples. The pipeline then identified a Conservative Peak Set (CPS) per age group, which consists of shared reproducible peak areas between at least two true replicates of the same age group; the CPSs include 20,004, 18,279, and 17,180 peaks for young, mid, and old samples respectively. We adjusted each genomic area in each of the peak sets to be of length 500 bp by limiting it to 250 bp around the summit of the peak, using a custom script. Then, the new peak sets of the three age groups were merged into one master 'bed' file using HOMER utility Mergepeaks (*Heinz et al., 2010*). A Heatmap plot was produced using BEDTools computeMatrix and plotHeatmap utilites.

### Bulk RNA-seq analysis

Read preprocessing, alignment and counting was performed by HTSeqGenie (4.12.0). Reads were aligned against the reference *Drosophila* genome (BDGP6), using gene models based on the Flybase *Drosophila melanogaster* Annotation Set (v6.02). Reads falling within coding sequences were counted to determine an estimate of gene expression. Counts were normalized using the TMM method, and then transformed using the voom method. Differential expression analysis was performed using the limma package (3.38.3). Differential peaks (adjusted p-value<0.05) were identified by limma (3.38.3) R package. PCA plot was performed on normalized counts (cpm) using PCAtools (1.2.0) R package. Volcano plots were generated using EnhancedVolcano (1.4.0) R package.

### scRNA-seq analysis

Single-cell RNA-seq data for three independent biological replicates were processed with cellranger count (Cell Ranger 3.0.2; 10x Genomics) using standard parameters and supplying a custom reference package based on *Drosophila melanogaster* reference genome BDGP6. Sequencing libraries with total UMI below 200 and above 2000, less than 500 genes expressed and more than 8% of mitochondrial genes detected were removed to eliminate potential empty droplets, cell aggregates and fragmented cells. A total of 4880 young, 1440 mid-aged and 5829 old cells were analyzed. Subsequent data analysis including normalization, batch correction, clustering and trajectory analysis was carried out with the Monocle 3 (0.2.1) R package. Automatic cell annotation was performed by garnett (0.1.17) R package. Differential genes in each cluster have been calculated based on the area under the ROC curve (AUC) using presto (1.0.0) R package. AUC was used to rank the genes for GSEA carried out with fgsea (1.12.0) R package. Gene sets were obtained from MsigDB (H, C2, C3, C5) (https://www.gsea-msigdb.org/gsea/msigdb/index.jsp). Custom gene sets were used as indicated in the Results section of the manuscript. Gene sets with adjusted p-value<0.05 were considered as significantly enriched.

### Post-translational histone modifications panel (active motif)

For measuring post-translational histone modifications (PTMs) during aging, whole midguts of $W^{Dah}$; *esg-Gal4, UAS-2xEYFP; Su(H)GBE-Gal80* flies were dissected at young (6-8d) and old (59-65d) timepoints. 100 midguts were used per sample, and there were two samples per age group. Midguts

were dissected in 1xPBS, transferred to a tube, the PBS was removed and the midgut tissue was snap frozen in liquid nitrogen and stored at −80°C until samples were shipped to Active Motif for analysis. In short, the following was performed by Active Motif: Two young and two old midgut samples, each a pool of 50, were frozen at −80°C upon receipt. A. Frozen samples were suspended in 250 μl of a sucrose based hypotonic buffer and homogenized using a battery powered plastic pestle, after which the volume of hypotonic buffer was increased to 1 ml and samples incubated for 30 min on ice. Nuclei were pelleted and washed with 500 μl hypotonic buffer prior to acid extraction (100 ul volume for two hours at 4°C). Cellular debris was removed by centrifugation, and the lysate used immediately as a 1.43 serial dilution in the multiplex assay containing all the histone H3 PTM beads.

Samples were tested in duplicate using four amounts/sample. Data sets with equivalent H3 Total signals were selected for downstream analysis. Net median fluorescent Intensity (Net MFI) associated with each PTM-specific bead was expressed as a ratio relative to Histone H3 Total signals for each well. Ratio values were averaged for each sample input amount (two technical replicates) and percent change in the ratio relative to the reference samples was determined. Statistical significance was calculated by using a student's t-test.

## Immunohistochemistry and imaging

Whole midguts from adult female flies were dissected in 1xPBS and fixed in a fixative solution (4% formaldehyde in a pH 7.5 solution containing 100 mM glutamic acid, 25 mM KCl, 20 mM MgSO4, 4 mM sodium phosphate dibasic, 1 mM MgCl2) for 45 min. Alternately, guts were incubated for 10 min after which the fixative solution was removed and replaced by 100% Methanol for 5 min followed by a 2 min incubation in a 1:1 solution of Methanol:1xPBS and 2 min in 1xPBS. The samples were then washed in wash buffer (1× PBS, 0.5% BSA and 0.1% Triton X-100) for 10 min at RT, and then for 1 h at 4°C on a rocker. Primary antibodies were diluted in wash buffer and samples were incubated overnight (O/N) at 4°C. Samples were washed in wash buffer for 1 hr at 4°C after which secondary antibodies (diluted in wash buffer) were added, along with Hoechst to label DNA and samples were incubated at RT for 2 hr. Samples were washed in wash buffer for 1 hr at 4°C and mounted in MoWiol mounting solution. The following primary antibodies were used in this study: mouse anti-Prospero (MR1A, DSHB, 1:250); rabbit anti-H3K27me2 (Abcam, ab24684, 1:500); chicken anti-GFP (Abcam, ab13970, 1:2000); rabbit anti-phospho-histone H3 (Sigma-Aldrich, 06–570, 1:1000). All secondary antibodies were obtained from Jackson ImmunoResearch Laboratories. Hoechst 3342 (Thermo Fisher Scientific, 62249) was used to label DNA. All imaging was performed on a Leica SP5 or SP8 confocal microscope and processed using FIJI is just Image J (FIJI), Adobe Photoshop and Illustrator.

## Image and statistical analyses

Quantification of cell numbers as well as fluorescence quantification was performed using FIJI (*Schindelin et al., 2012*). Equal areas of posterior midguts were analyzed in control and experimental samples. Statistical analyses were performed in Graphpad Prism 7. Significance in two-condition experiments was evaluated by student's t-test (two-tailed, parametric). Multiple condition experiments were evaluated by an ordinary one-way ANOVA, with either Dunnett's post-hoc to compare a control group with experimental conditions or Tukey's post-hoc when all conditions were compared to each other. Partek Flow Genomic Analysis Software was used for Cluster five analysis and image generation in *Figure 2D, E and F*. Partek Flow was also used to generate all graphs illustrating single gene expression across indicated conditions.

## Acknowledgements

We thank Rebeccah Riley for initial sequencing ATAC-seq samples at the Buck Institute for Research on Aging. We thank Nadja Katheder for help with axenic fly cultures and Max Adrian for help with image analysis. We also thank all core facilities including the Buck Inst. FACS core as well as the Genentech FACS core, Next Generation Sequencing core and Imaging Core for all their help. We are grateful for resources available from the Bloomington *Drosophila* Stock Center (Indiana University) and the Vienna *Drosophila* Resource Center. This work was funded in part by the EMBO Long-Term Fellowship ALTF 1516–2011.

# Additional information

## Competing interests

Helen M Tauc, Imilce A Rodriguez-Fernandez, Jason A Hackney, Subhra Chaudhuri, Zora Modrusan: employee of Genentech Inc. The other authors declare that no competing interests exist.

## Funding

| Funder | Grant reference number | Author |
|---|---|---|
| EMBO | ALTF 1516-2011 | Jerome Korzelius |
| European Research Council | AdG 268515 | Bruce A Edgar |
| NIH | GM124434 | Bruce A Edgar |

The funders had no role in study design, data collection and interpretation, or the decision to submit the work for publication.

## Author contributions

Helen M Tauc, Conceptualization, Data curation, Formal analysis, Investigation, Methodology, Writing - original draft, Project administration, Writing - review and editing; Imilce A Rodriguez-Fernandez, Investigation, Methodology, Collaboration on single cell RNA-seq experiments; Jason A Hackney, Data curation, Software, Bioinformatics analyses of sequencing data; Michal Pawlak, Data curation, Software, Visualization, Bioinformatics analyses of sequencing data; Tal Ronnen Oron, Software, Formal analysis, Visualization, Bioinformatics analyses of sequencing data; Jerome Korzelius, Hagar F Moussa, Formal analysis, Investigation, Trithorax experiments; Subhra Chaudhuri, Investigation, Library preparation and sequencing of ATAC- and RNA-seq experiments; Zora Modrusan, Supervision, Investigation, Library preparation and sequencing of ATAC- and RNA-seq experiments; Bruce A Edgar, Supervision, Funding acquisition; Heinrich Jasper, Conceptualization, Data curation, Supervision, Project administration, Writing - review and editing

## Author ORCIDs

Helen M Tauc (iD) https://orcid.org/0000-0002-0694-2387
Imilce A Rodriguez-Fernandez (iD) http://orcid.org/0000-0002-5112-4834
Hagar F Moussa (iD) https://orcid.org/0000-0003-3463-0126
Bruce A Edgar (iD) http://orcid.org/0000-0002-3383-2044
Heinrich Jasper (iD) https://orcid.org/0000-0002-6014-4343

## Decision letter and Author response

Decision letter https://doi.org/10.7554/eLife.62250.sa1
Author response https://doi.org/10.7554/eLife.62250.sa2

# Additional files

## Supplementary files

• Supplementary file 1. Table of results from ATAC-seq of aging ISCs. Related to *Figure 1*. Differential peak analysis between aged time-points (mid-age, old and geriatric) and young ISCs. Tabs separate age comparisons and significantly regulated peaks.

• Supplementary file 2. Table of results from bulk RNA-seq of aging ISCs. Related to *Figure 1*. Differential gene expression analysis between aged time-points (mid-age, old and geriatric) and young ISCs. Tabs separate age comparisons and significantly regulated genes.

• Supplementary file 3. Table illustrating age-associated genes regulated at RNA and chromatin level. Related to *Figure 1*. Significantly differentially regulated genes at mRNA and chromatin levels across aged time-points in ISCs.

- Supplementary file 4. Table outlining genes regulated in old ISCs are upregulated in old Cluster 4 cells. Related to *Figures 1* and *2*. Differentially regulated genes in old ISCs are also found in gene trajectory modules that are upregulated in Cluster 4. See also *Figure 2—figure supplement 5*.

- Supplementary file 5. Table of results from Pc-RNAi and trx-RNAi RNA-seq. Related to *Figure 5*. Differential gene expression analysis after Pc-RNAi and trx-RNAi. Tabs separate Pc-RNAi and trx-RNAi comparisons as well as significantly regulated genes.

- Supplementary file 6. Table of Pc-RNAi ATAC-seq differential peak analysis. Related to *Figure 5*. Significantly differentially regulated peak in promoter regions after Pc-RNAi in ISCs.

- Transparent reporting form

## Data availability

Data generated and analysed are included in the manuscript, figures and figure supplements. All sequencing data generated in this study have been deposited in GEO under accession code GSE164317 and GSE157796.

The following datasets were generated:

| Author(s) | Year | Dataset title | Dataset URL | Database and Identifier |
|---|---|---|---|---|
| Tauc H, Ronnen-Oron T | 2021 | Age-related changes in Pc gene regulation disrupt lineage fidelity in intestinal stem cells | https://www.ncbi.nlm.nih.gov/geo/query/acc.cgi?acc=GSE164317 | NCBI Gene Expression Omnibus, GSE164317 |
| Pawlak M | 2021 | Aging ISCs and Polycomb KD ISCs | https://www.ncbi.nlm.nih.gov/geo/query/acc.cgi?acc=GSE157796 | NCBI Gene Expression Omnibus, GSE157796 |

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
