## [Decision Letter]

**Acceptance summary:**

How age-related changes in somatic stem cells impact lineage decisions is an important topic in aging. Using genome-wide chromatin accessibility and transcriptome analysis and single cell RNA-seq to explore this in *Drosophila* intestine stem cell, this study found that *Pc*-mediated chromatin regulation is central regulators of lineage identity bias during aging.

**Decision letter after peer review:**

Thank you for submitting your article "Age-related changes in Polycomb gene regulation disrupt lineage fidelity in intestinal stem cells" for consideration by *eLife*. Your article has been reviewed by three peer reviewers, and the evaluation has been overseen by a Reviewing Editor and Jessica Tyler as the Senior Editor. The reviewers have opted to remain anonymous.

The reviewers have discussed the reviews with one another and the Reviewing Editor has drafted this decision to help you prepare a revised submission.

Essential Revisions (as detailed below in more detail):

1) Integrative analysis of omics data, e.g., ATAC-seq and RNA-seq, or RNAseq and scRNA-seq, or three of them;

2) Add rescue experiments for most of knockdown and RNAi experiment;

3) Other additional data to support the logics between central claims and omics data.

We also suggest that you try to address the additional comments in the reviews, but these are not essential.

Reviewer #1:

The paper by Tauc et al. uses the *Drosophila* intestine to investigate age-related changes in ISC lineage fidelity. The authors show that ISCs exhibit transcriptomic and chromatin accessibility changes through deregulating of Polycomb target genes and activating *JNK*, which contributing to dysplasia in aging flies. This is an important finding which may lead to effective perturbation of PcG gene regulation of the age-related loss in tissue homeostasis. However, some results need to be presented with more details and rigorously.

1) Figure 1A. the number of flies in mid-age, old, geriatric groups only have 3 for ATAC-seq and RNA-seq. It would be nice to increase the number of flies in these groups to 6 and the same to the young group. Moreover, it's surprising to see 7 flies in the young group in Figure 1—figure supplement 1.

2) The number of cells present in Figure 2A is not shown and not compared with the estimated numbers from literature, which presents the quality of the ATAC-seq and RNA-seq.

3) Figure 3B why there is no mid-age group to quantify the proportion of EEs and EE progenitors cells?

4) Figure 4A There is a mislabeling in the top of relative H3K27me27. Since H3K27me1 and H3K27me2 are also deposited in different chromatin domains to activate transcription, however, the H3K27me1 changes are missing in Figure 4—figure supplement 1. Also, there is no statistics analysis in Figure 4—figure supplement 1. No scale bar in Figure 4C and Figure 6—figure supplement 1.

5) Did the author investigate expression changes of polycomb repressive complex 2, such as *Pc* or E(z) in aging flies?

Reviewer #2:

The authors found transcriptomic and chromatin accessibility changes prime ISCs toward the EE lineage in aging flies, related with Polycomb genes. Then they confirmed the necessity of *Pc* for EE specification, and long-term *Pc* knock-down in aging also reduced dysplasia levels. Finally, they found commensal dysbiosis-mediated *JNK* activation causes EE increases.

This work is rigorous and well-grounded, elucidating intrinsic changes in the aging ISCs and how these changes impact ISC lineage fidelity, further, age-related pathologies. They covered Polycomb genes as key regulators of ISC lineage identity, providing a fundamental understanding of aging.

1) A numbered summary of any substantive concerns:

a) In chromatin accessibility analysis on aging ISCs, the authors performed a motif analysis on differentially accessible promoter regions, and found an enrichment for Trl and z motifs. Are there differences between up-regulated promoter regions and down-regulated ones?

b) As the majority of old ISCs reside in an activated cell state (~50%), and the "stressed ISC" population makes up a small percentage (<10%), whether activated or stressed ISCs have relation with EE specification, further affect tissue homeostasis?

c) After knocking down *Pc*, there is a decrease in accessibility of ISC genes and increased accessibility included homeobox TFs, while ISC proliferation was not affected. How do the authors explain this apparent discrepancy?

And what about EE genes? Is there a decrease in accessibility of EE genes after knocking down *Pc* in ISCs?

d) For the comment that "while the increase in EE cell numbers was repeatedly and robustly observed in animals that exhibited age-related epithelial dysplasia, we noticed that in animals that exhibited reduced dysplasia, the increase in EE cells was less pronounced", please explain it more clearly. (It seems no data shown to support the comment)

e) The increase of H3K27me3 from young to old flies is 24%, does the H3K27me3 signal exhibit broader coverage and increased intensity at transcriptional start sites, like HSC and muscle SC?

Reviewer #3:

In this perspective, Tauc et al. preformed genome-wide chromatin accessibility and transcriptome analysis as well as single cell RNA-seq to explore stem cell- intrinsic changes in the aging *Drosophila* intestine. They found *Pc*-mediated chromatin regulation can determine the EE cell specification and ratio in the aged *Drosophila* intestine. Given somatic stem cell is a critical model for regeneration and aging field, the incentive for such a well-compiled comparative study is obvious. However, the deliverables that this ambitious plan eventually offered are relatively weak.

First, the authors picked the cherry of Polycomb (*Pc*) solely used an excuse from motif analysis of ATAC-seq, which largely lower their value of key data from RNA-seq and Single-cell analyses. What are the envisions from enrichment of DNA repair and TNFalpha/NFkB signaling from Single-cell sequencing, and enrichment of glutathione metabolic process and drug metabolism from RNA-seq ? Did they regulated by *Pc* as well or response for increase of EE cells ? An upregulation of neuronal genes was also found by the authors and their previous study, why this has not been considered by functional assay. And, though the author presented a series of RNAi and knockdown assay to show the effects, there is no single rescue experiment was conducted in parallel.

Second, at the end of single cell section, the authors claimed that scRNA-seq data revealed that the upregulation of EE genes in aged ISCs observed by bulk RNA-seq. This will cause a further question if this particular case of EE-genes could be extrapolated to other clusters of cells and pathways, not to mentions that ATAC-seq signatures were not integrated. The author could give a clear description of how many DEGs in RNA-seq that could validated by scRNA-seq? in which clusters, the sc-RNA is most consistent with RNA, did they correlated with cell numbers? How much variation of expression of Polycomb target genes could explained by shift of promoters peaks ?

Third, the method and parameters used in the study are unclear, with only the package name was given in most case. And, it looks like the author used different methods to map and identify DEGs in Bulk RNA-seq and RNAi experiments. There is clear and huge evidence to show that different pipelines and combinations of software can get a completely different output. So, it is really weird distinct methods were used for two sets of RNA-seq analyses.

After all, the major difficulty in analyzing such a dataset lies in the well-define strategies for cross-omics data. Although there have been some international joint-ventures to systematically catalogue the signatures and changes. The amount of method generated so far are still far from comprehensive and requires huge amounts of comparisons. To enable this dataset to make meaningful predictions, novel informatic algorithms was encouraged to be designed in this study which would make full use of generated resources.

---

## [Author Response]

Essential Revisions (as detailed below in more detail):1) Integrative analysis of omics data, e.g., ATAC-seq and RNA-seq, or RNAseq and scRNA-seq,or three of them;

We agree that further analysis of the generated datasets will provide additional hypotheses to test. Such analysis can go into many different directions, and we anticipate it to continue by our and other groups once our data have been made public. Our original manuscript provided a first example for the power and usefulness of such analysis and of the generated data: the combined analysis of the omics dataset resulted in an unexpected hypothesis, namely that aging causes a shift towards EE cell specification in the ISC lineage, and that changes in *Pc* gene regulation contribute to that process. Strikingly, our genetic studies testing this hypothesis succeeded in supporting a new model for how aging impacts stem cell function in the intestine. We believe that this work thus highlights the strength of the omics analysis and of its combination with functional genetic studies. The value of additional “integrative” analysis of omics data was thus not immediately obvious to us, but we have now performed the following additional analysis to address the reviewers’ concerns, which we believe add substantial support to our original findings:

1) A one-to-one comparison of the ATAC-seq and RNA-seq data from aging ISCs that examines the correlation of transcriptional changes with changes in promoter chromatin accessibility between the different age groups. We have added a supplementary figure to Figure 1 that shows these comparisons (Figure 1—figure supplement 2). Interestingly, while changes in gene expression and promoter accessibility do not strongly correlate when comparing all differentially expressed genes with all differentially open chromatin peaks (A), a strong positive correlation is observed when this analysis is performed for genes that significantly (p<0.05) change in both transcription and promoter accessibility (B). Strikingly, of these differentially accessible promoters with significantly changed transcripts, 58% (18/31) contain a Trl and z motifs. We have added a table outlining these genes (Table 3, Supplementary file 3).

This analysis thus further confirms the enrichment of *Pc*-binding motifs in the promoters that are differentially regulated in aging ISCs. We have added and clarified this in the text [Results].

2) A “pseudo bulk” transcriptome analysis was performed on the combined ISC Clusters 1, 3 and 4 in our scRNA-seq and compared with our bulk RNAseq from sorted ISCs. This analysis shows a significant correlation of differentially expressed genes when comparing young with mid-aged and young with old samples (correlation is weaker in the second comparison, potentially due to an age-related increase of transcriptional noise). We have added this analysis as new Figure 2—figure supplement 2, Results.

3) To understand which clusters contribute most to the age-associated changes we observed in our bulk RNA-seq, we performed pseudo bulk analysis and GSEA on each individual cluster and compared differences between mid-aged vs. young and old vs. young cells. Genes that were significantly up- or down-regulated in our bulk RNA-seq (for either the mid vs. young and old vs. young comparisons) were used for GSEA. We compiled this analysis into new Figure 2—figure supplement 3. This analysis shows that the differentially regulated genes found in our bulk RNA-seq at mid and old ages are also changed at corresponding timepoints in our scRNA-seq. Importantly, the analysis shows that these changes are most significant in the ISC and progenitor cell clusters (#1-4), not differentiated cell types (Clusters #5-7). These results confirm the robustness and complementarity of both our bulk and scRNA-seq datasets. We have clarified this in the main text [Results].

4) We would also like to re-iterate the findings we had already reported in the original submission, namely that the gene expression changes observed in the bulk RNA-seq likely reflect the acquisition of an “old” ISC state from a “young” ISC state. As we originally described in the main text [Results], we performed trajectory analyses from Cluster 1 (the young ISCs) to Cluster 4 (composed almost exclusively of old ISCs, Figure 2B) to identify transcriptome trajectories that are affected by aging. A significant portion of genes upregulated in old ISCs in our bulk RNA-seq were also identified in gene modules that show higher expression in Cluster 4 (represented by modules as shown in Figure 2—figure supplement 5). This analysis shows that a significant number of genes identified in the bulk RNA-seq are also identified in the scRNA-seq and most likely have functional relevance in driving the transition of a “young” ISC into an “old” ISC state.

Overall, these results confirm that the bulk RNA-seq data (which provide more robust and deep quantification of the transcriptome) and the scRNA-seq data (which of course allow better resolution of cell lineages) complement each other. We have clarified this in the main text as described above and in the Discussion.

In addition to points described above, we have also added tables that better illustrate the changes we found in ATAC-seq, RNA-seq and scRNA-seq (Tables 1-6 in Supplementary files 1-6).

2) Add rescue experiments for most of knockdown and RNAi experiment;

The requested “rescue” experiments were not specified by the reviewers, and we have tried to respond to this request as best we can, by providing further technical validation of our knockdown experiments (A), and by performing genetic interaction studies to rescue phenotypes associated with specific knockdown experiments (B):

A) To validate our knockdown experiments technically, we have used standard approaches used in *Drosophila*: we have performed knock downs with distinct RNAi lines targeting the same gene and observed the same phenotypes, and have also confirmed phenotypes using germline loss of function alleles of the gene of interest:

– We used two different RNAi lines targeting the *Pc* gene and two targeting the E(z) gene (both *Pc* and E(z) are critical for canonical *Pc* complex function). All four RNAi lines produced the same phenotype supporting the conclusion that *Pc*/E(z) are required for EE cell differentiation.

– Knockdown or heterozygous loss of function mutants of *trx*, the antagonist of *Pc*, led to the opposite phenotype of *Pc* knockdown (more EEs), thus providing further support for our conclusion that *Pc* and *trx* complexes regulate EE differentiation.

In addition, the RNAi lines we used have been validated in previous studies.

**Table resptable1:** 

RNAI line	Reference
E(z)-RNAi BL36068	Abdusselamoglu et al., 2019
*Pc*-RNAi BL36070; E(z)-RNAi BL33659	Kang et al., 2015
*Trx*-RNAi (GD37715)	Smith et al., 2013
Puc-RNAi (GD3018)	Hu and Jasper, 2019

B) Another interpretation of the request to provide “rescue” experiment would be a phenotypic rescue. To address this, we performed the following experiments:

– The scute gene (an EE specification factor) was downregulated in our RNA-seq after *Pc* knockdown. To test whether overexpression of scute could rescue the loss of EE differentiation after *Pc* knockdown, we overexpressed scute and simultaneously knocked down *Pc* in the ISC lineage. This experiment showed that scute is able to rescue the loss of EEs after *Pc* depletion, thus confirming that it acts downstream of *Pc* (now reported in Figure 5).

– We generated transheterozygotes of the *trx* and *Pc* loss of function alleles *trx^E2^* and *Pc^1^* to ask whether reducing the gene dose of *Pc* would rescue the increase in EE cells observed in *trx* heterozygotes. Indeed, we observed such a rescue (now reported in Figure 5), further validating the results of our RNAi knockdown experiments.

3) Other additional data to support the logics between central claims and omics data.

We were a bit baffled by this request, as it doesn’t specify what exactly is expected. Our study identifies an age-related shift in ISC specification towards the EE fate. This shift was identified by our bulk RNA-seq analysis and supported by ATAC-seq and single cell RNA-seq analyses, which at the same time linked changes in *Pc*-regulated gene expression to this shift in EE specification. As far as we can tell, our findings and conclusions follow logically from the analysis of the omics datasets and were then also supported by genetic studies.

Nevertheless, to provide additional support for our conclusions, we have performed the additional bioinformatics analysis described above and have also performed the genetic interaction studies described above between *Pc* and scute and between *Pc* and *trx*. All these studies further support our central claims and we believe that the study is now significantly stronger.

We also suggest that you try to address the additional comments in the reviews, but these are not essential.Reviewer #1:The paper by Tauc et al. uses the *Drosophila* intestine to investigate age-related changes in ISC lineage fidelity. The authors show that ISCs exhibit transcriptomic and chromatin accessibility changes through deregulating of Polycomb target genes and activating JNK, which contributing to dysplasia in aging flies. This is an important finding which may lead to effective perturbation of PcG gene regulation of the age-related loss in tissue homeostasis. However, some results need to be presented with more details and rigorously.1) Figure 1A. the number of flies in mid-age, old, geriatric groups only have 3 for ATAC-seq and RNA-seq. It would be nice to increase the number of flies in these groups to 6 and the same to the young group. Moreover, it's surprising to see 7 flies in the young group in Figure 1—figure supplement 1.

We apologize for this confusion. Rather than using only 3 flies for each group, in these experiments, each n represents a group of ~120 flies that were manually dissected, and whose intestines were dissociated and FACS sorted. The number of flies used is described in the figure legends and in the Materials and methods section, and we have now clarified this in the text [Results].

2) The number of cells present in Figure 2A is not shown and not compared with the estimated numbers from literature, which presents the quality of the ATAC-seq and RNA-seq.

We thank the reviewer for pointing this out. We have added the number of cells analyzed to the main text [Results] as well as to the Materials and methods section. Of note, Figure 2A is a separate single-cell RNA-seq experiment and does not reflect the quality of the bulk ATAC-seq or RNA-seq experiments described in Figure 1.

3) Figure 3B why there is no mid-age group to quantify the proportion of EEs and EE progenitors cells?

Figure 3B is a confirmation experiment to support the conclusions drawn from Figure 3A and as such we had refrained from including another group of labor-intensive dissections to this analysis. The question addressed in 3B is whether the age-dependent increase in Pros+ cells is observed independent of the genetic background, and analysis young and old cohorts is sufficient to answer that question.

4) Figure 4A There is a mislabeling in the top of relative H3K27me27. Since H3K27me1 and H3K27me2 are also deposited in different chromatin domains to activate transcription, however, the H3K27me1 changes are missing in Figure 4—figure supplement 1.

We thank the reviewer for noting this typo – we have corrected the label from H3K27me27 to H3K27me2. H3K27me1 was unfortunately not a modification included in the panel of antibodies used for this analysis.

Also, there is no statistics analysis in Figure 4—figure supplement 1.

We thank the reviewer for pointing out the lack of statistics. We have added p-Values to the table. P-Values were calculated by a t-test for two biological replicates and two technical replicates per biological replicate.

No scale bar in Figure 4C and Figure 6—figure supplement 1.

We thank the reviewer for pointing out the lack of scale bars. We have added them to both figures.

5) Did the author investigate expression changes of polycomb repressive complex 2, such as Pc or E(z) in aging flies?

The expression levels of *Pc* complex genes such as *Pc* and E(z) as well as *trx* complex genes were not significantly changed on a transcriptional level during aging. We did not investigate the binding of these complexes and speculate that aging most likely affects post-transcriptional regulation and/or function. We have now mentioned this in the Discussion.

Reviewer #2:The authors found transcriptomic and chromatin accessibility changes prime ISCs toward the EE lineage in aging flies, related with Polycomb genes. Then they confirmed the necessity of Pc for EE specification, and long-term Pc knock-down in aging also reduced dysplasia levels. Finally, they found commensal dysbiosis-mediated JNK activation causes EE increases.This work is rigorous and well-grounded, elucidating intrinsic changes in the aging ISCs and how these changes impact ISC lineage fidelity, further, age-related pathologies. They covered Polycomb genes as key regulators of ISC lineage identity, providing a fundamental understanding of aging.1) A numbered summary of any substantive concerns:a) In chromatin accessibility analysis on aging ISCs, the authors performed a motif analysis on differentially accessible promoter regions, and found an enrichment for Trl and z motifs. Are there differences between up-regulated promoter regions and down-regulated ones?

The reviewer brings up an interesting point. Promoters containing Trl and z motifs that were more accessible in old ISCs did not show GO term or pathway enrichment. Less accessible promoters were enriched for homeobox protein domain-containing genes. More than half the genes that were significantly regulated on both the mRNA and chromatin accessibility level (common genes in Figure 1—figure supplement 2B and Table 3, Supplementary file 3) contained a Trl and z motif. These data support the conclusion that misregulation of *Pc*/*trx* contributes to mis-regulation of genes and potentially promoter accessibility. We did not further explore the function of the genes containing Trl/z motifs as that would be beyond the scope of this paper.

b) As the majority of old ISCs reside in an activated cell state (~50%), and the "stressed ISC" population makes up a small percentage (<10%), whether activated or stressed ISCs have relation with EE specification, further affect tissue homeostasis?

The results from our single cell studies suggests that indeed, only a small population of the ISCs becomes “stressed” or “damaged”. This is a novel finding and suggests that the majority of ISCs maintain a robust transcriptional profile even during aging. We show two clusters (#4 and #8) that we defined as stressed or damaged ISCs. Cluster #8 appears to be a separate population of damaged ISCs that may indicate damage beyond repair or senescence based on pathway enrichment analysis. Cluster #4 cells are still transcriptionally closer to activated ISCs. The relationship between stressed or activated ISCs and EE differentiation remains unclear, but our scRNA-seq data suggest that stressed or activated ISCs and EE-specified ISCs can be defined as separate populations. Future studies to elucidate the relationship between these various populations will be fascinating.

c) After knocking down Pc, there is a decrease in accessibility of ISC genes and increased accessibility included homeobox TFs, while ISC proliferation was not affected. How do the authors explain this apparent discrepancy?

We apologize that our description was not clear enough. We do in fact not believe that there is a discrepancy between these findings. We find that *Pc* knockdown in ISCs does not have a significant effect on ISC proliferation and, accordingly, we did not observe a change in accessibility of promoters at proliferation genes. Rather than affecting proliferation, it is possible that the differential accessibility of promoters at the ISC genes esg and spdo as well as the homeobox genes Lim1, Abd-B, oc and CG4328, contributes to shifts in cell identity and lineage specification. Further studies will be needed to dissect the contribution of each of these genes to the aging phenotype.

And what about EE genes? Is there a decrease in accessibility of EE genes after knocking down Pc in ISCs?

We did not observe an obvious decrease in accessibility of known EE genes after knocking down *Pc*. But we did observe a decrease in the mRNA expression of critical EE specification genes, such as *ase*, *sc* and *phyl*, suggesting that critical EE specification genes are regulated on a level that is independent of their promoter chromatin accessibility. We have clarified this in the text [Results].

d) For the comment that "while the increase in EE cell numbers was repeatedly and robustly observed in animals that exhibited age-related epithelial dysplasia, we noticed that in animals that exhibited reduced dysplasia, the increase in EE cells was less pronounced", please explain it more clearly. (It seems no data shown to support the comment)

We apologize for the lack of clarity of this statement. We have changed this sentence to the following in the text:

“the increase in EE cell numbers was robustly and consistently observed in animals that exhibited age-related epithelial dysplasia. In old animals that did not exhibit dysplasia, however, the increase in EE cells was less pronounced.”

Additionally, Figure 6—figure supplement 1 shows data on how dysplasia levels correlate with EE numbers. Note that even in some aged intestines, low dysplasia levels correlate with low EE number.

e) The increase of H3K27me3 from young to old flies is 24%, does the H3K27me3 signal exhibit broader coverage and increased intensity at transcriptional start sites, like HSC and muscle SC?

This is a very interesting question that we cannot answer yet. In this report, we only focused on overall H3K27me2/3 signal in whole midgut preparations and did not pursue ChIP-seq as the material we could obtain from ISCs is very limited.

Reviewer #3:In this perspective, Tauc et al. preformed genome-wide chromatin accessibility and transcriptome analysis as well as single cell RNA-seq to explore stem cell- intrinsic changes in the aging *Drosophila* intestine. They found Pc-mediated chromatin regulation can determine the EE cell specification and ratio in the aged *Drosophila* intestine. Given somatic stem cell is a critical model for regeneration and aging field, the incentive for such a well-compiled comparative study is obvious. However, the deliverables that this ambitious plan eventually offered are relatively weak.

We thank the reviewer for characterizing our study as well-compiled and would like to respectfully disagree that the results we present are “relatively weak”. We do in fact find an unexpected shift in lineage specification in the ISC lineage of aging flies, and provide evidence that changes in *Pc* gene regulation causes this shift. These findings are a direct consequence of the combination of RNAseq, ATACseq and scRNAseq approaches and would not have been possible without these. We have now added additional genetic interaction studies to probe this model further.

First, the authors picked the cherry of Polycomb (Pc) solely used an excuse from motif analysis of ATAC-seq, which largely lower their value of key data from RNA-seq and Single-cell analyses. What are the envisions from enrichment of DNA repair and TNFalpha/NFkB signaling from Single-cell sequencing, and enrichment of glutathione metabolic process and drug metabolism from RNA-seq ?

We respectfully disagree that the findings that changes in *Pc* gene regulation contribute to the age-related shift in lineage specification are of “low value”. The biological insight here is critical and novel, and strongly supported by the data: The motif analysis clearly indicates enrichment for Trl and z motifs in differentially accessible promoters in aged ISCs. Furthermore, Abd-B and Ubx (classic Polycomb targets) are among the top upregulated genes in aged ISCs, strongly suggesting a role for Polycomb during aging. We have edited the text to better highlight the importance of these findings (Results). As described above (Response to Essential revisions point 2), we have now also added additional data from genetic interaction experiments that support a role for *Pc* in EE lineage specification (Figure 5).

The increase in glutathione metabolic processes as well as DNA repair and TNFalpha/NFkB signaling in aged ISCs have been previously reported. The well-established inflammatory environment in old midguts contributes to elevated oxidative stress and an accumulation of reactive oxygen species in ISCs, further exacerbated by the loss of the master antioxidant response regulator, CncC/Nrf2 (Hochmuth et al., 2011). Increased oxidative stress is linked to increases in DNA damage in ISCs (Park et al., 2012). Thus, the enrichment for these processes is not novel and reflects what has already been shown in the literature. We have now better explained and discussed this in the text [Results].

Did they regulated by Pc as well or response for increase of EE cells ?

The wording of the question is confusing, but we ascertained that it asked whether the upregulation/enrichment of DNA damage/glutathione metabolism in old ISCs is a response to the increase in EEs. We have not addressed the consequences of higher EE numbers on intestinal homeostasis, however, a previous study reported that overproduction of EEs results in higher ISC proliferation and decreased lifespan (Amcheslavsky et al., 2014). Our studies indicate that higher JNK activity promotes EE differentiation and higher EE numbers may promote ISC proliferation and tissue dysplasia.

An upregulation of neuronal genes was also found by the authors and their previous study, why this has not been considered by functional assay. And, though the author presented a series of RNAi and knockdown assay to show the effects, there is no single rescue experiment was conducted in parallel.

The upregulation of neuronal genes most likely reflects the increase in EE cell differentiation, as many genes expressed in neurons are also expressed in EEs (Guo et al., 2019; Hartenstein, Takashima and Adams, 2010).

As to the second part of the question, we have discussed this in the common response above, and now have included the following rescue experiments:

Concomitant overexpression of wild-type *sc* with *Pc*-RNAi that showing that *sc* is able to rescue the loss of EE after *Pc* knockdown. Combining *Pc* and *Trx* loss of function alleles shows that loss of *Pc* can rescue the loss of *Trx*-mediated increase in EE specification (Figure 5).

Second, at the end of single cell section, the authors claimed that scRNA-seq data revealed that the upregulation of EE genes in aged ISCs observed by bulk RNA-seq. This will cause a further question if this particular case of EE-genes could be extrapolated to other clusters of cells and pathways, not to mentions that ATAC-seq signatures were not integrated. The author could give a clear description of how many DEGs in RNA-seq that could validated by scRNA-seq? in which clusters, the sc-RNA is most consistent with RNA, did they correlated with cell numbers?

Please refer to our answer to point 1 of the Essential revisions requested above.

How much variation of expression of Polycomb target genes could explained by shift of promoters peaks ?

A strong positive correlation is observed for genes that significantly (p<0.05) change in both transcription and promoter accessibility (Figure 1—figure supplement 2B). Strikingly, of these differentially accessible promoters with significantly changed transcripts, 58% (18/31) contain a Trl and z motifs.

Third, the method and parameters used in the study are unclear, with only the package name was given in most case. And, it looks the author used different methods to map and identify DEGs in Bulk RNA-seq and RNAi experiments. There is clear and huge evidence to show that different pipelines and combinations of software can get a completely different output. So, it is really weird distinct methods were used for two sets of RNA-seq analyses.

We thank the reviewer for this comment and have re-done the analyses using the same pipeline and have updated Figure 5 and the text to reflect the new analysis. Differentially expressed genes after *Pc*RNAi or *Trx*-RNAi were overall consistent with our previous analysis: loss of *Pc* resulted in upregulation of proteolytic genes and a decrease in key EE specification genes such as *ase*. A minor discrepancy was observed in the statistical significance for some genes, such as some proteolytic genes as well as *sc* and *phyl*. The expression of these genes, however, is consistently increased (for the proteolytic genes) and decrease (for *sc* and *phyl*) upon *Pc* knockdown. We now have added graphs to (Figure 5C and added Figure 5—figure supplement 2) to illustrate the actual rpkm values and up- and down-regulation of these genes. The new analysis consistently showed that loss of *trx* results in an upregulation of EE genes and cell cycle genes. Furthermore, GSEA shows a positive enrichment for GO terms such as central nervous system neuron differentiation, peptide hormone secretion and insulin secretion, highly indicative of EE characteristics. The upregulation of key EE genes after *trx*-RNAi and a downregulation upon *Pc*-RNAi consistently supports the conclusion that PcG and *trx*G act antagonistically in the ISC decision between EE and EC fate. Ultimately, the conclusions from these experiments remain consistent with our previous conclusions.

After all, the major difficulty in analyzing such a dataset lies in the well-define strategies for cross-omics data. Although there have been some international joint-ventures to systematically catalogue the signatures and changes. The amount of method generated so far are still far from comprehensive and requires huge amounts of comparisons. To enable this dataset to make meaningful predictions, novel informatic algorithms was encouraged to be designed in this study which would make full use of generated resources.

We agree that further analysis of the generated datasets will be very interesting. Our manuscript highlights one conclusion that we believe is biologically meaningful and we have performed experiments to test the emerging hypothesis with rigorous genetic analysis. As pointed out by the other reviewers, these findings are interesting and important, and we do not think that we would have arrived at these conclusions without the transcriptomics and epigenomic analysis presented here. Since all the raw data generated will be posted online, we look forward to additional analysis performed on these datasets with new algorithms that may provide new hypotheses to be tested in the future.

References:

Abdusselamoglu, M. D., Landskron, L., Bowman, S. K., Eroglu, E., Burkard, T., Kingston, R. E., and Knoblich, J. A. (2019). Dynamics of activating and repressive histone modifications in *Drosophila* neural stem cell lineages and brain tumors. Development, 146(23). doi:10.1242/dev.183400

Guo, X., Yin, C., Yang, F., Zhang, Y., Huang, H., Wang, J.,... Xi, R. (2019). The Cellular Diversity and Transcription Factor Code of *Drosophila* Enteroendocrine Cells. Cell Rep, 29(12), 4172-4185 e4175. doi:10.1016/j.celrep.2019.11.048

Hartenstein, V., Takashima, S., and Adams, K. L. (2010). Conserved genetic pathways controlling the development of the diffuse endocrine system in vertebrates and *Drosophila*. Gen Comp Endocrinol, 166(3), 462-469. doi:10.1016/j.ygcen.2009.12.002

Hu, D. J., and Jasper, H. (2019). Control of Intestinal Cell Fate by Dynamic Mitotic Spindle Repositioning Influences Epithelial Homeostasis and Longevity. Cell Rep, 28(11), 2807-2823 e2805.

doi:10.1016/j.celrep.2019.08.014

Kang, H., McElroy, K. A., Jung, Y. L., Alekseyenko, A. A., Zee, B. M., Park, P. J., and Kuroda, M. I. (2015). Sex comb on midleg (Scm) is a functional link between PcG-repressive complexes in *Drosophila*. Genes Dev, 29(11), 1136-1150. doi:10.1101/gad.260562.115

Park, J. S., Lee, S. H., Na, H. J., Pyo, J. H., Kim, Y. S., and Yoo, M. A. (2012). Age- and oxidative stressinduced DNA damage in *Drosophila* intestinal stem cells as marked by Gamma-H2AX. Exp Gerontol, 47(5), 401-405. doi:10.1016/j.exger.2012.02.007

Smith, H. F., Roberts, M. A., Nguyen, H. Q., Peterson, M., Hartl, T. A., Wang, X. J.,... Bosco, G. (2013). Maintenance of interphase chromosome compaction and homolog pairing in *Drosophila* is regulated by the condensin cap-h2 and its partner Mrg15. Genetics, 195(1), 127-146. doi:10.1534/genetics.113.153544